# Age, Genesis and Tectonic Setting of the Sayashk Tin Deposit in the East Junggar Region: Constraints from Lu–Hf Isotopes, Zircon U–Pb and Molybdenite Re–Os Dating

Zhenjun Sun [1,2], Guanghu Liu [1,2,*], Yunsheng Ren [1,2], Xi Chen [3], Xinhao Sun [4] , Chengyang Wang [1,2] and Zuowu Li [5]

[1] School of Earth Sciences, Institute of Disaster Prevention, Sanhe 065201, China
[2] Hebei Key Laboratory of Earthquake Dynamics, Sanhe 065201, China
[3] No.1 Bureau of China Metallurgical Geology Bureau, Sanhe 065201, China
[4] Chengdu Center of China Geological Survey, Chengdu 610081, China
[5] Xinjiang Branch of China National Geological Exploration Center of Building Materials Industry, Urumqi 830092, China
* Correspondence: liuguanghu@cidp.edu.cn

**Abstract:** The Sayashk tin (Sn) deposit is located within the southern part of the Eastern Junggar orogenic belt in Xinjiang Province and forms part of the Kalamaili alkaline granite belt. There are many Sn polymetallic deposits in the area. To constrain the age, genesis, and tectonic setting of the Sayashk tin deposit in the East Junggar region, we conducted a bulk-rock geochemical analysis of the granite porphyry (SR1) and medium- to fine-grained granite (SR2) hosts of the deposit, LA-ICP-MS zircon U–Pb dating and Lu–Hf c Re–OS dating and combined our results with the metallogenic conditions and other geological characteristics of the deposit. The results show that the Sayashk Sn deposit is indeed spatially, temporally, and genetically closely related to the granite porphyry and medium-fine-grained granite. Both zircon U–Pb ages are 308.2 ± 1.5 Ma and 310.9 ± 1.5 Ma, respectively. The isochron age of molybdenite is 301.4 ± 6.7 Ma, which represents the crystallization age of the granite porphyry and medium-fine-grained granite. Therefore, all of them formed in the late Carboniferous epoch. The medium-fine-grained granites and granite porphyry are characteristically rich in Si and alkali, poor in Ca and Mg, rich in high field-strength elements (HFSE, e.g., Zr, Hf) and Ce, and deficient in Ba, Sr, Eu, P, and Ti. They are typical A-type granites, showing the characteristics of a mixed crustal mantle source. The $\varepsilon Hf(t)$ values of the zircon from the granite porphyry (SR1) range from 10.27 to 16.17 (average 13.71), $\varepsilon Hf(t)$ values of the zircon from the medium-fine-grained granites (SR2) are between 5.72 and 9.21 (average 7.08), and the single model ages ($T_{DM1}$) and two-stage model ages ($T_{DM2}$) of the granite porphyry (SR1) fall within the ranges of 319~535 Ma and 339~644 Ma. The single model ages ($T_{DM1}$) and two-stage model ages ($T_{DM2}$) of the medium-fine-grained granites (SR2) fall within the ranges of 346~479 Ma and 309~557 Ma. There is little difference between their two-stage model ages and zircon U–Pb ages, indicating that the Sayashk granite may be the product of partial melting of juvenile crustal. Combined with previous research results, the Sayashk Sn deposit formed in a post-collision extensional tectonic setting after the late Carboniferous in the Kalamaili area.

**Keywords:** alkaline granite; zircon U–Pb geochronology; zircon Hf isotope compositions; molybdenite Re–Os dating; Sayashk tin deposit; east Junggar region

## 1. Introduction

The Beilekuduk tin (Sn) ore belt in the eastern Junggar region of northern Xinjiang is the first metallogenic belt dominated by Sn found. There are a number of Sn deposits (occurrences) distributed from the west to the east, such as Kamst, Ganliangzi, Beilekuduk,

Sayashk, Sujiquan, and Hongtujingzi in the region, which have long been studied by mineral deposit scientists locally and globally [1–5]. The eastern Junggar area is an important tin and gold metallogenic belt in northern Xinjiang. Tin mineralization is related to late Paleozoic granite [6]. Many researchers have carried out a large number of studies on petrography, geochemistry, isotope chronology, ore genesis, and ore-forming fluid properties and typical Sn deposits all over the world [1–3,7–11]. It is believed that the granites related to mineralization are mainly S-type granites originating from the partial melting of crustal materials [12–14]. However, an increasing amount of research data shows that most granites related to Sn mineralization in the world are closely related to highly differentiated granites in time and space, as well as genesis [15–19]. Generally, granites related to Sn mineralization are biotite granite, two-mica granite, potassium feldspar granite, quartz porphyry, granite porphyry, alkali feldspar granite, and other granitic rocks, and most of them belong to highly differentiated S-type, A-type, and I-type granites [20–29]. They are generally characterized by high concentrations of Si and alkali, enrichment of incompatible elements B, F, Rb, Th, U, low concentrations of elements such as Ca, Mg, Fe, Ti, Ba, Sr, Zr, and strong negative Eu anomalies, indicating that the magma has undergone a strong crystallization differentiation process [11,15].

Many researchers have studied the petrogenetic and metallogenic ages of tin deposits in the Belkuduk Sn ore belt with different dating techniques. The petrogenetic and metallogenic ages are mainly concentrated at 341–275 Ma [6,13,14,30]. The large span of petrogenetic and metallogenic ages, the diversity of testing methods, and the accuracy of testing have seriously restricted further understanding of the metallogenic regularity of tin deposits and the effect of prospecting and prediction in the eastern Junggar area. The Sabei granite is the first A-type granite discovered to develop Sn deposits in China. Similar to other rocks in the Belkuduk Sn ore belt, it has attracted extensive attention [2,8,31–33]. However, as a granite porphyry closely related to mineralization, it has not been given as much attention as the alkaline granite developed in the mining area, and there are few relevant studies. Meanwhile, a lot of researchers hold different views on its petrogenetic and metallogenic age; for example, Zhang et al. (1992) dated the single zircon U–Pb of the Sayashk amphibole granite to 290 ± 11 Ma [34], Yang (2010) obtained a weighted average age of 277 ± 11 Ma by using LA-ICP-MS zircon U–Pb [8], and Lin et al. (2007) obtained two weighted average ages of 313 ± 2 Ma and 314 ± 5 Ma by using Shrimp zircon U–Pb dating [31]. Lin et al. (2008) thought that the Sn mineralization age was 324.2 ± 3.4 Ma by using Shrimp zircon U–Pb dating of cassiterite quartz veins in the Sayashk Sn deposit [30].

To date, the metallogenic age of the Sayashk Sn deposit has not been well resolved. We carried out geochemical element analysis of granite and granite porphyry, LA-ICP-MS zircon U–Pb dating, Lu–Hf isotope composition, and molybdenite Re–Os dating to precisely determine the formation age of the Sayashk tin deposit, the types and genesis of granites related to mineralization, and the nature of the magma source area; we will use this information to discuss the geodynamic background of its formation. This study will provide an important scientific basis for further understanding the mineralization in this area.

## 2. Regional Geology

The eastern Junggar Orogenic Belt in northern Xinjiang is located in the northern part of the eastern Tianshan mountains (Figure 1a). It is an important part of the Central Asian orogenic belt. It is bounded by the Kalamaili fault in the south and the Erqisi fault in the north [35–37]. The strong tectonic-magmatic activity formed rich mineral resources. The Kalamaili area is located in the south of the East Junggar Orogenic Belt. A series of alkaline and ca-alkaline granite intrusions, such as the Yemaquan pluton, Laoyaquan pluton, Beilekuduk pluton, Sabei pluton, and Huangyangshan pluton, are developed on the north side of the Kalamaili deep fault and are distributed along the NW direction in the area. The Kamust Sn deposit, Ganliangzi Sn deposit, Hongtujingzi Sn deposit, Beilekuduke Sn deposit, Sayashk Sn deposit, Huangyangshan graphite deposit

and Sujiquan Sn deposit (graphite deposit) are distributed in the NW direction along the Kalamaili deep fault [2,3,9,38–40].

Devonian and Carboniferous strata are mainly developed in the region (Figure 1b). The lithology is mainly tuffaceous siltstone and pyroclastic rock mixed with limestone and carbonaceous marlstone, which are exposed along both sides of the Kalamaili deep fault.

A small number of Silurian strata is sporadically exposed south of the Kalamaili deep fault. Regionally, WNW-trending faults are developed. The Kalamaili deep fault is a typical deep fault. In addition, Suji Qingshui and Kupu Kubusu deep faults roughly parallel to it are also developed [41]. They jointly control the emplacement of alkaline granitoids after the collisional orogeny, accompanied by the intrusion of a series of slightly alkaline granite porphyry and Sn mineralization [31].

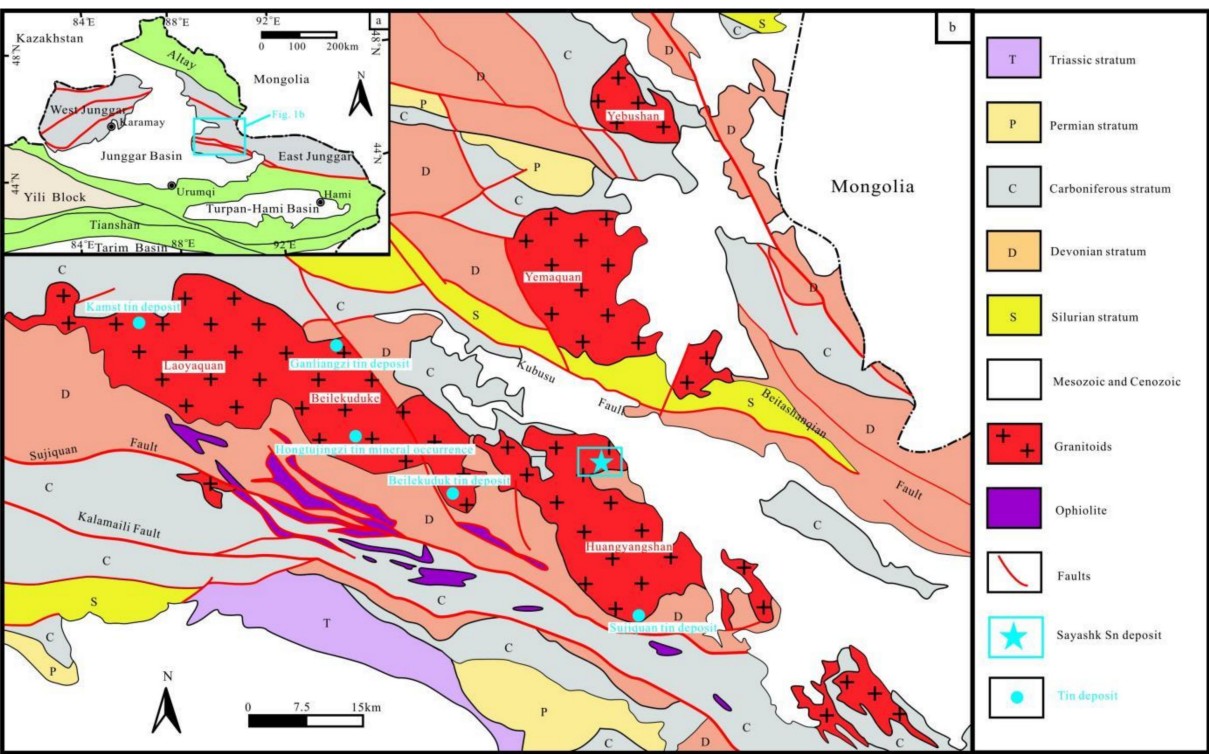

**Figure 1.** (**a**) Geological map of Xinjiang (modified after [42]). (**b**) Simplified geological map of the Kalamaili area, eastern Junggar (modified after [43]).

The late Carboniferous magmatic activity in the area was intense, forming large-scale and complex intrusive rocks such as Laoyaquan, Beilekuduk, Yemaquan, Sabei, and Huangyangshan, all of which occur as batholiths. Granitic magma is characterized by multicycle evolution, and its composition generally shows calc-alkaline-sub-alkaline-alkaline evolution, which is conducive to the differentiation and enrichment of Sn. The Sabei composite alkaline granite is closely related to the Sayashk Sn deposit and is located in the southeastern part of Mount Kalamaili and on the north side of the Huangyangshan pluton [9]. The Sabei composite alkaline granite is a typical rock in the Kalamali alkaline granite belt, and it intrudes into the tuffaceous sandstone and siltstone of the Beitashan Formation of the middle Devonian and the Jiangbastao Formation of the lower Carboniferous. Due to the multistage pulsating intrusion, the intrusion of the rocks from early to late is fine-grained riebeckite granite-medium fine-grained arfvedsonite granite-medium coarse-grained arfvedsonite granite-porphyritic riebeckite granite-riebeckite granite porphyry (dyke), which formed by four successive pulsation intrusions of homologous magma [32,33].

### 3. Geology of the Ore Deposit

The Carboniferous intrusive rocks in the Sayashk Sn deposit are widely distributed. The main lithologies are pink medium fine-grained porphyritic syenogranite and potassium feldspar granite. The medium fine-grained porphyritic syenogranite is further named medium fine-grained arfvedsonite alkaline granite. The dykes are dominated by quartz albite porphyry, granite porphyry, quartz porphyry, granite aplite, quartz vein, and other hypabyssal plutons. The strike of the dykes is NE and NNE. The fault structures in the mining area are mainly distributed in the NW, NEE or NNE direction and are generally small in scale. Many researchers generally believed that the magmatic rock closely related to mineralization was granite [2,31–33,44]. After detailed field geological surveys and exploration reports, we found that mineralization was closely related to quartz veins and granite porphyry or quartz albite porphyry veins (Figure 2). The ore body is a Sn mineralized quartz vein with obvious greisenization, silicification, epidotization, malachitization, covellite and other Cu mineralization and molybdenization. With the ore body as the center, alteration occurs to both sides, and the alteration intensity decreases with increasing distance to both sides. There is no obvious boundary between the surrounding rock and the altered rock, showing a gradual transition relationship, granite → granite porphyry → Sn-containing quartz veins.

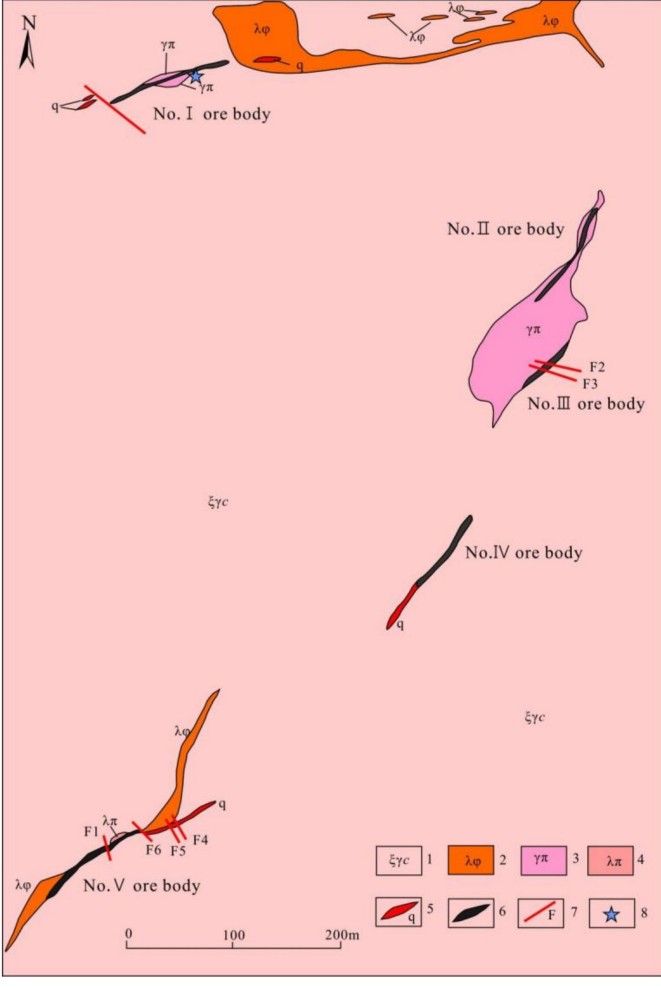

**Figure 2.** Geological map of Sayashk Sn mining area. 1—medium fine-grained alkaline granite; 2—quartz albite porphyry; 3—granite porphyry; 4—quartz porphyry; 5—quartz vein; 6—tin ore body; 7—fault; 8—sampling location.

Five ore bodies are preliminarily delineated in the deposit, of which the I and II ore bodies are large in scale and have strong tin mineralization.

Ore body I: the ore body is approximately 160 m long and 1–2 m wide, and the occurrence is 190° ∠ 85°. It is a quartz vein Sn ore body hosted in granite porphyry, with obvious silicification. The cassiterite and molybdenite are distributed in a dense star point, malachitization is distributed in a thin film.

Ore body II: the ore body is approximately 100 m long and 1–2 m wide, and the occurrence is 190° ∠ 85°. It occurs in granite porphyry with obvious silicification. The cassiterite and molybdenite are distributed in a dense star point, malachitization is distributed in a thin film.

Ore (mineralized) bodies III, IV, V: each ore body is approximately 50–100 m long and 0.8–1 m wide, and its occurrence is about 235° ∠ 78°. The ore bodies occur in granite porphyry or quartz albite porphyry with obvious silicification. The cassiterite and molybdenite are distributed in a dense star point, malachitization is distributed in a thin film. Sn grade is 1.0%–1.5%, Mo content is 0.001%–0.05%, and Cu content is 0.01%–0.10%.

## 4. Samples and Analytical Procedures

### 4.1. Samples Descriptions

The granite and granite porphyry samples (Figure 3) closely related to mineralization, as well as molybdenite samples, were collected from the ore body I of the Sayashk Sn deposit. The sampling coordinates are 90°21′29″ east longitude and 45°10′56″ north latitude. The weathered surface and fresh surface of the medium-fine-grained granite are gray-white, with a medium fine-grained structure. The particle size is mainly distributed between 1–3 mm, and the massive structure is mainly composed of alkali feldspar (65%), plagioclase (5%), quartz (25%), and a small number of dark minerals (biotite + amphibole accounts for approximately 5%) (Figure 3a,d,e). The weathered surface of the granite porphyry is yellowish brown, and the fresh surface is dark gray with a porphyritic structure. The phenocrysts are mainly alkali feldspar (5%) and quartz (5%), and the particle size is mainly between 2–4 mm. The matrix has cryptocrystalline and vein structures. Molybdenite mainly occurs in disseminated, micro vein, and film forms (Figure 3b,f,g).

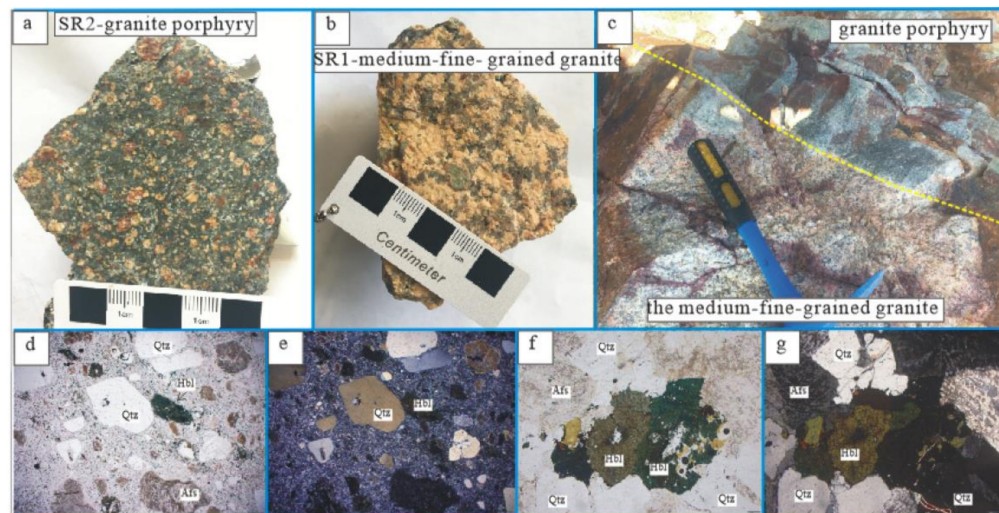

**Figure 3.** Filed photos, hand specimens and photomicrographs of typical rock units and ores from the Sayashk Sn deposit. (**a**) Granite porphyry with a grayish white color and a porphyritic structure; (**b**) the medium-fine-grained granite with a grayish white color and a medium-fine-grained structure; (**c**) intrusive contact relationship between granite and granite porphyry; (**d**,**e**) granite porphyry under single polarizer and orthogonal polarizer, respectively; (**f**,**g**) medium-fine-grained granite under single polarizer and orthogonal polarizer, respectively.

### 4.2. Analytical Procedures

Major and trace element geochemical analyses were undertaken at Yanduzhongshi Geological Analysis Laboratories Ltd. Major oxide concentrations were measured with an X-ray fluorescence (XRF) spectrometer (Shimadzu, Kyushu, Japan) and the trace element concentrations were determined by ICP-MS techniques (Analytik Jena AG, Jena, Germany). The analytical precisions estimated through repeated analyses of standards were better than ±0.01% for XRF and ±5% for ICP-MS analysis. The detailed analytical methods and procedures are described in Liu et al. (2008) [45].

Data testing and analysis of zircon U–Pb dating were performed at Yanduzhongshi Geological Analysis Laboratories Ltd. using LA-ICP-MS. M90 ICP-MS (Shimadzu, Columbia, MD, USA)was used for the analysis with a New Wave UP213 laser ablation system (Thermo Fisher Scientific, Carlsbad, CA, USA). During the sample tests, the diameter of the laser ablation was 30 μm, SRM610 was used as the external standard, and Si was used as the internal standard to calculate the trace element contents of the zircons. Zircon standard 91,500 was used as the external standard for isotope fractionation correction, and for every 5–10 sample points analyzed, two standard sample analyses and one Plesovice analysis were conducted for monitoring. U–Pb dating was performed using the ICPMS Data Cal 4.3 program (ICPMS Data Cal 4.3, Wuhan, China) [46]. The U–Pb age calculation and concordia plots were processed using ISOPLOT 3.0 [47]. The common Pb correction process was performed according to the method proposed by Andersen [48].

Zircon Lu–Hf isotope analysis was carried out in-situ by using an NWR193 laser-ablation microprobe (Elemental Scientific Lasers LLC, Bozeman, MT, USA), attached to a Neptune multicollector ICP-MS at Yanduzhongshi Geological Analysis Laboratories Ltd. Instrumental conditions and data acquisition were comprehensively described by Wu et al. (2006) [49]. A stationary spot was used for the present analyses, with a beam diameter of 35 μm. Helium was used as carrier gas to transport the ablated sample from the laser-ablation cell to the ICP-MS torch via a mixing chamber mixed with Argon. In order to correct the isobaric interferences of $^{176}Lu$ and $^{176}Yb$ on $^{176}Hf$, $^{176}Lu/^{175}Lu = 0.02658$ and $^{176}Yb/^{173}Yb = 0.796218$ ratios were determined [50]. For instrumental mass bias correction Yb isotope ratios were normalized to $^{172}Yb/^{173}Yb$ of 1.35274 [50] and Hf isotope ratios to $^{179}Hf/^{177}Hf$ of 0.7325 using an exponential law. The mass bias behavior of Lu was assumed to follow that of Yb, mass bias correction protocols details were described by Wu et al. (2006) [49]. Zircon 91,500 and Plesovice were used as the reference standards during our routine analyses, Initial $^{176}Hf/^{177}Hf$ ratios εHf (t) were calculated with reference to the chondritic reservoir (CHUR) of Blichert-Toft and Albarede (1997) [51] at the time of zircon growth from the magma. The single-stage Hf model age (TDM1) is calculated relative to the depleted mantle with present-day $^{176}Hf/^{177}Hf = 0.28325$ and $^{176}Lu/^{177}Hf = 0.0384$ [52] (e.g., Griffin et al., 2000).

Re–Os isotope analyses were completed on the separated samples at the Re–Os Laboratory of the National Research Center of Geoanalysis, Chinese Academy of Geological Sciences. Details of the chemical separation procedure are described by Shirey and Walker (1995) [53], Stein et al. (1997) [54], Du et al. (2004) [55], Xie et al. (2007) [56], and Mao et al. (1999) [57]. The Re–Os model age was calculated as $t = 1/\lambda [\ln(1 + {}^{187}Os/{}^{187}Re)]$, where $\lambda$ is the decay constant of $^{187}Re$, and the adopted value is $\lambda = 1.666 \times 10^{-11}$ year$^{-1}$. Positive and inverse isochrons were computed with the ISOPLOT program [47].

## 5. Results

### 5.1. Whole-Rock Major and Trace Elements

Whole-rock major and trace element compositions of five medium-fine-grained granite samples (defined SR2) and five granite porphyry samples (defined SR1) are presented in Table 1. These samples generally have loss on ignition (LOI) values of <2 wt.%, suggesting that they have not been affected by alteration. Overall, all ten samples are characterized by high concentrations of silica and alkalis and low concentrations in oxides of Al, Mg, and Ca, and the concentration of $K_2O$ is greater than the concentration of $Na_2O$. However,

five medium-fine-grained granite samples (sample No. SR-2-1~SR-2-5) on average contain $w$ ($SiO_2$) = 78.65% wt.%, $w$ ($Al_2O_3$) = 11.10%, $w$ (CaO) = 0.22%, $w$ (MgO) = 0.074%, and $w$ ($Na_2O$) + w ($K_2O$) = 8.25%. Five granite porphyry samples (sample No. SR-1-1~SR-1-5) contain $w$ ($SiO_2$) = 71.27%, $w$ ($Al_2O_3$) = 13.27%, $w$ (CaO) = 0.72%, $w$ (MgO) = 0.22%, and $w$ ($Na_2O$) + w ($K_2O$) = 9.60%.

**Table 1.** Major (%) and trace (ppm) element compositions of the granite porphyry and the medium-fine-grained granite from the Qiushuwan deposit.

| Sample | SR-1-1 | SR-1-2 | SR-1-3 | SR-1-4 | SR-1-5 | SR-2-1 | SR-2-2 | SR-2-3 | SR-2-4 | SR-2-5 |
|---|---|---|---|---|---|---|---|---|---|---|
| $SiO_2$ | 71.88 | 71.26 | 72.49 | 73.57 | 72.16 | 79.01 | 78.77 | 78.16 | 79.23 | 78.09 |
| $TiO_2$ | 0.29 | 0.31 | 0.26 | 0.23 | 0.29 | 0.08 | 0.07 | 0.10 | 0.08 | 0.06 |
| $Al_2O_3$ | 13.32 | 13.49 | 13.33 | 12.91 | 13.31 | 11.07 | 11.02 | 11.26 | 10.77 | 11.37 |
| $TFe_2O_3$ | 3.05 | 3.15 | 2.85 | 2.71 | 3.12 | 1.20 | 1.23 | 1.42 | 1.27 | 1.12 |
| CaO | 0.85 | 0.89 | 0.80 | 0.71 | 0.84 | 0.18 | 0.19 | 0.20 | 0.20 | 0.31 |
| MgO | 0.23 | 0.23 | 0.20 | 0.21 | 0.23 | 0.07 | 0.08 | 0.08 | 0.07 | 0.07 |
| MnO | 0.06 | 0.06 | 0.07 | 0.03 | 0.04 | 0.02 | 0.02 | 0.02 | 0.02 | 0.01 |
| $K_2O$ | 4.94 | 5.04 | 5.12 | 4.62 | 5.16 | 4.78 | 4.06 | 4.58 | 4.17 | 4.30 |
| $Na_2O$ | 4.67 | 4.70 | 4.56 | 4.73 | 4.47 | 3.54 | 3.99 | 3.89 | 3.80 | 4.14 |
| $P_2O_5$ | 0.07 | 0.07 | 0.06 | 0.06 | 0.07 | 0.03 | 0.03 | 0.03 | 0.03 | 0.03 |
| LOI | 0.35 | 0.35 | 0.30 | 0.34 | 0.38 | 0.29 | 0.21 | 0.25 | 0.16 | 0.30 |
| Total | 99.70 | 99.55 | 100.05 | 100.13 | 100.07 | 100.24 | 99.67 | 99.99 | 99.78 | 99.80 |
| AR | 4.87 | 4.78 | 4.65 | 5.55 | 4.43 | 4.40 | 5.92 | 5.24 | 5.50 | 5.87 |
| A/CNK | 0.91 | 0.91 | 0.92 | 0.92 | 0.92 | 0.98 | 0.97 | 0.96 | 0.97 | 0.94 |
| A/NK | 1.02 | 1.02 | 1.02 | 1.01 | 1.03 | 1.01 | 1.01 | 0.99 | 1.00 | 0.99 |
| Rb | 199.19 | 190.78 | 212.20 | 187.24 | 218.03 | 200.28 | 170.64 | 184.36 | 179.84 | 157.69 |
| Sr | 59.70 | 56.35 | 59.55 | 51.46 | 52.62 | 10.17 | 6.52 | 6.20 | 7.30 | 6.10 |
| Y | 68.23 | 69.28 | 73.89 | 67.67 | 72.76 | 52.65 | 51.24 | 56.58 | 52.67 | 58.56 |
| Zr | 633.83 | 654.34 | 671.58 | 612.52 | 625.19 | 293.81 | 315.98 | 280.93 | 312.97 | 289.08 |
| Nb | 14.87 | 14.11 | 14.99 | 15.66 | 14.55 | 11.92 | 12.10 | 15.16 | 12.26 | 14.03 |
| Cd | 0.60 | 0.61 | 0.64 | 0.48 | 0.73 | 0.53 | 0.38 | 0.50 | 0.35 | 0.39 |
| In | 0.17 | 0.17 | 0.17 | 0.17 | 0.18 | 0.15 | 0.15 | 0.15 | 0.14 | 0.11 |
| Cs | 11.14 | 11.12 | 11.51 | 10.00 | 15.15 | 5.88 | 5.98 | 5.27 | 8.90 | 2.98 |
| Ba | 494.93 | 508.19 | 472.93 | 361.18 | 508.23 | 34.19 | 23.57 | 19.77 | 24.70 | 15.55 |
| La | 39.92 | 43.78 | 51.17 | 26.16 | 43.65 | 36.73 | 37.16 | 35.14 | 39.20 | 32.03 |
| Ce | 84.06 | 100.94 | 108.03 | 56.62 | 100.63 | 74.06 | 72.56 | 71.96 | 71.17 | 69.23 |
| Pr | 10.66 | 12.13 | 13.62 | 7.82 | 12.39 | 9.57 | 9.81 | 9.21 | 10.48 | 8.49 |
| Nd | 45.14 | 51.43 | 56.18 | 32.74 | 52.16 | 38.94 | 40.15 | 37.89 | 42.35 | 33.98 |
| Sm | 11.79 | 12.66 | 13.63 | 9.69 | 12.75 | 10.19 | 10.53 | 10.23 | 10.65 | 9.65 |
| Eu | 0.54 | 0.58 | 0.54 | 0.36 | 0.56 | 0.01 | 0.00 | 0.00 | 0.00 | 0.01 |
| Gd | 12.23 | 12.95 | 13.89 | 10.16 | 13.09 | 10.09 | 9.97 | 9.71 | 9.77 | 9.67 |
| Tb | 2.07 | 2.18 | 2.26 | 1.92 | 2.16 | 1.69 | 1.69 | 1.73 | 1.63 | 1.75 |
| Dy | 12.99 | 12.87 | 13.73 | 12.19 | 13.21 | 9.94 | 9.96 | 10.62 | 9.96 | 10.75 |
| Ho | 2.78 | 2.78 | 2.92 | 2.75 | 2.78 | 2.12 | 2.16 | 2.35 | 2.20 | 2.37 |
| Er | 7.85 | 7.66 | 8.17 | 7.92 | 7.83 | 5.95 | 6.15 | 6.66 | 6.26 | 6.73 |
| Tm | 1.08 | 1.03 | 1.16 | 1.12 | 1.06 | 0.78 | 0.84 | 0.91 | 0.85 | 0.92 |
| Yb | 7.71 | 7.28 | 7.68 | 8.15 | 7.38 | 5.14 | 5.78 | 6.36 | 5.77 | 6.05 |
| Lu | 1.09 | 1.07 | 1.15 | 1.17 | 1.07 | 0.74 | 0.85 | 0.86 | 0.83 | 0.84 |
| Hf | 18.32 | 18.07 | 19.09 | 18.28 | 17.34 | 10.78 | 11.49 | 10.73 | 10.21 | 10.58 |
| Ta | 1.18 | 1.12 | 1.07 | 1.11 | 1.07 | 0.89 | 0.89 | 1.02 | 0.84 | 0.92 |
| Tl | 0.67 | 0.64 | 0.69 | 0.64 | 0.68 | 0.56 | 0.45 | 0.46 | 0.52 | 0.49 |
| Th | 10.74 | 9.93 | 11.23 | 11.40 | 10.23 | 11.76 | 12.01 | 12.17 | 11.87 | 12.23 |
| U | 3.97 | 3.55 | 4.11 | 4.22 | 3.36 | 3.87 | 3.80 | 5.15 | 3.19 | 3.36 |
| ΣREE | 239.91 | 269.34 | 294.13 | 178.77 | 270.72 | 205.95 | 207.61 | 203.63 | 211.12 | 192.48 |
| LREE | 192.11 | 221.52 | 243.17 | 133.39 | 222.14 | 169.50 | 170.21 | 164.43 | 173.85 | 153.39 |
| HREE | 47.80 | 47.82 | 50.96 | 45.38 | 48.58 | 36.45 | 37.40 | 39.20 | 37.27 | 39.08 |
| LREE/HREE | 4.02 | 4.63 | 4.77 | 2.94 | 4.57 | 4.65 | 4.55 | 4.19 | 4.66 | 3.92 |
| (La/Yb$_N$) | 3.71 | 4.31 | 4.78 | 2.30 | 4.24 | 5.13 | 4.61 | 3.96 | 4.87 | 3.80 |
| Nb/Ta | 38.25 | 45.79 | 52.51 | 29.37 | 48.75 | 43.77 | 45.01 | 37.08 | 50.13 | 15.26 |
| Zr/Hf | 34.59 | 36.2 | 35.19 | 33.51 | 36.05 | 27.25 | 27.51 | 26.18 | 30.65 | 27.32 |
| δEu | 0.14 | 0.14 | 0.12 | 0.11 | 0.13 | 0.0029 | 0.98 | 0.99 | 0.98 | 0.0021 |

Note: A/CNK = molar Al/(Ca + Na + K); A/NK = molar Al/(Na + K); δEu = 2 EuN/(SmN × GdN).

In the $SiO_2$ vs. $K_2O$ + $Na_2O$ discrimination diagrams, all ten samples plot in the granite field, which is consistent with the identification results of hand specimens (Figure 4a). As the samples have high $SiO_2$ contents (>70 wt.%), we used an AR vs. $SiO_2$ diagram for classification (Figure 4b). The ten samples plot in the alkaline series, whereas the enclaves

generally plot in the alkaline series. In the A/CNK versus A/NK diagram, all samples are metaluminous (Figure 4c).

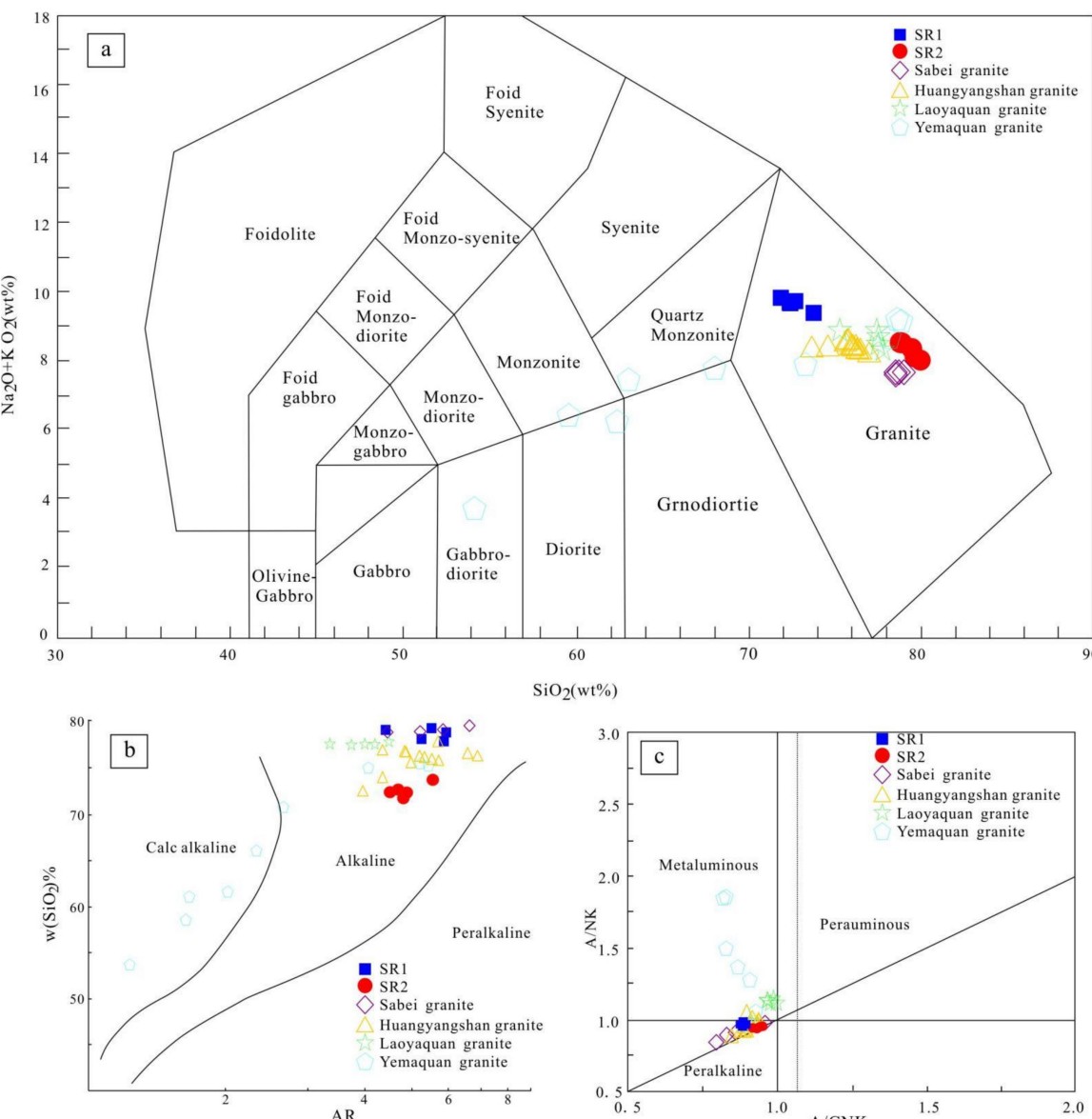

**Figure 4.** Geochemical characteristics of the medium-fine-grained granites and granite porphyry in the Sayashk Sn deposit. (**a**) SiO$_2$ vs. (Na$_2$O + K$_2$O) diagram. (**b**) AR vs. SiO$_2$ diagram. (**c**) A/CNK vs. A/NK diagram. Sabei granite after [30], Huangyangshan granite after [58], Laoyaquan granite after [9], Yemaquan granite after [59].

The total rare earth elements (REEs) of the medium-fine-grained granites and granite porphyry range from 192.48 ppm to 211.12 ppm and 178.77 ppm to 270.72 ppm, respectively. On a chondrite-normalized REE diagram (Figure 5a), the two granite distribution curves have a similar trend, showing a slightly right-leaning "seagull" shape. The fractionation of LREE and HREE is not obvious, and there is an obvious negative Eu anomaly. (La/Yb)$_N$ ratio SR2 = 3.80–5.13, (La/Yb)$_N$ ratio SR1 = 2.30–4.78.

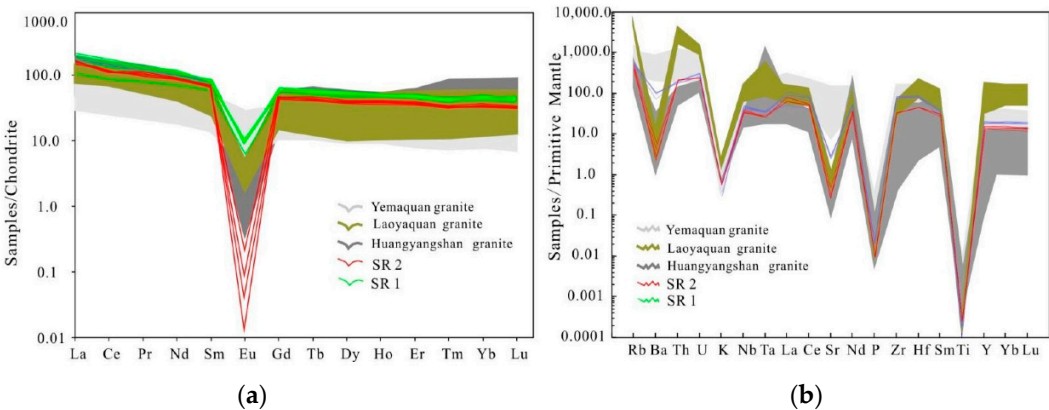

**Figure 5.** (**a**) Primitive mantle-normalized trace element and (**b**) chondrite-normalized REE patterns of the SR2 and SR1 in the Sayashk Sn deposit. Huangyangshan granite after [58], Laoyaquan granite after [9], Yemaquan granite after [59].

The granite porphyry and granite in the Sayashk deposit are enriched in LILEs, such as Rb, Th, and Nd, and HFSEs, such as Zrand Hf, but depleted in elements such as Ba, Sr, K, P, Ti, and Eu (Figure 5b). Both are characterized by significantly negative Eu anomalies, suggesting dominant plagioclase fractionation in more evolved melts.

### 5.2. CL Imaging and U–Pb Geochronology

Zircon CL images of the medium-fine-grained granites and granite porphyry are shown in Figure 6. Most of the zircons in the SR1 sample are colorless and transparent, and some are dark, while most of the zircons in the SR2 sample are dark, and both zircons are long biconical. The lengths of the zircons range from 100~200 μm. Most of them are typical magmatic zircons characterized by euhedral crystals with oscillatory zoning or linear zoning, which represents the crystallization of the magma. The zircon U–Pb isotope results (Table 2) show that the Th average content of the samples SR1 is 155.00 ppm, the U average content is 346.84 ppm, the Th/U average ratio is 0.42. In sample SR2, the average value of Th is 165.01 ppm, the average value of U is 388.42 ppm, and the Th/U average ratio is 0.42. Both samples have high Th/U ratios (>0.4) and belong to typical magmatic zircons [60]. Both Th and U contents show a good positive correlation, which is consistent with the characteristics of typical magmatic zircons [61–63].

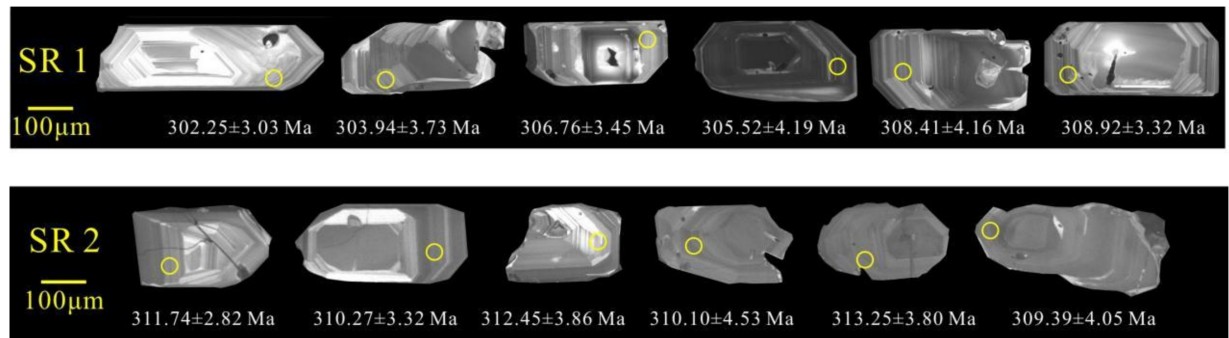

**Figure 6.** Part of zircon CL images showing the locations of LA-ICP-MS measurement spots and associated $^{206}Pb/^{238}U$ ages for the SR2 and SR1 in the Sayashk Sn deposit.

**Table 2.** Results of the zircon LA-ICP-MS U–Pb dating results for SR2 and SR1 in the Sayashk district.

| Analysis Point | U (ppm) | Th (ppm) | Th/U | Isotopic Ratios | | | | | | Ages (Ma) | | | | | |
|---|---|---|---|---|---|---|---|---|---|---|---|---|---|---|---|
| | | | | $^{207}$Pb/$^{206}$Pb | 1σ | $^{207}$Pb/$^{235}$U | 1σ | $^{206}$Pb/$^{238}$U | 1σ | $^{207}$Pb/$^{206}$Pb | 1σ | $^{207}$Pb/$^{235}$U | 1σ | $^{206}$Pb/$^{238}$U | 1σ |
| SR-2-01 | 344 | 140 | 0.41 | 0.0532 | 0.0012 | 0.3619 | 0.0072 | 0.0495 | 0.0005 | 335.30 | 49.49 | 313.66 | 5.39 | 311.74 | 2.82 |
| SR-2-03 | 380 | 150 | 0.40 | 0.0532 | 0.0013 | 0.3614 | 0.0090 | 0.0493 | 0.0005 | 336.80 | 56.31 | 313.23 | 6.70 | 310.27 | 3.32 |
| SR-2-04 | 380 | 141 | 0.37 | 0.0541 | 0.0013 | 0.3711 | 0.0096 | 0.0497 | 0.0006 | 376.94 | 54.88 | 320.48 | 7.09 | 312.45 | 3.86 |
| SR-2-05 | 348 | 148 | 0.43 | 0.0532 | 0.0014 | 0.3663 | 0.0109 | 0.0497 | 0.0007 | 338.24 | 58.49 | 316.91 | 8.13 | 312.68 | 4.16 |
| SR-2-06 | 342 | 136 | 0.40 | 0.0526 | 0.0016 | 0.3578 | 0.0109 | 0.0493 | 0.0007 | 312.45 | 69.24 | 310.54 | 8.18 | 310.10 | 4.53 |
| SR-2-09 | 306 | 118 | 0.39 | 0.0521 | 0.0011 | 0.3515 | 0.0080 | 0.0490 | 0.0005 | 291.54 | 48.94 | 305.85 | 6.00 | 308.17 | 3.07 |
| SR-2-11 | 389 | 163 | 0.42 | 0.0535 | 0.0015 | 0.3628 | 0.0104 | 0.0491 | 0.0006 | 351.79 | 63.11 | 314.27 | 7.77 | 309.26 | 3.48 |
| SR-2-12 | 399 | 145 | 0.36 | 0.0527 | 0.0016 | 0.3624 | 0.0114 | 0.0498 | 0.0007 | 314.04 | 68.20 | 313.98 | 8.49 | 313.40 | 4.13 |
| SR-2-13 | 386 | 192 | 0.50 | 0.0526 | 0.0010 | 0.3588 | 0.0076 | 0.0493 | 0.0005 | 312.55 | 43.41 | 311.31 | 5.69 | 310.35 | 3.05 |
| SR-2-14 | 410 | 171 | 0.42 | 0.0534 | 0.0013 | 0.3631 | 0.0091 | 0.0493 | 0.0007 | 347.15 | 54.79 | 314.56 | 6.75 | 310.42 | 3.99 |
| SR-2-15 | 361 | 163 | 0.45 | 0.0529 | 0.0012 | 0.3599 | 0.0086 | 0.0493 | 0.0005 | 324.27 | 50.07 | 312.17 | 6.43 | 309.99 | 3.18 |
| SR-2-16 | 293 | 99 | 0.34 | 0.0525 | 0.0013 | 0.3559 | 0.0093 | 0.0492 | 0.0007 | 305.60 | 55.43 | 309.16 | 6.93 | 309.39 | 4.05 |
| SR-2-17 | 321 | 140 | 0.44 | 0.0526 | 0.0013 | 0.3622 | 0.0088 | 0.0497 | 0.0007 | 310.68 | 58.43 | 313.87 | 6.55 | 312.92 | 4.28 |
| SR-2-19 | 690 | 365 | 0.53 | 0.0526 | 0.0010 | 0.3632 | 0.0073 | 0.0495 | 0.0006 | 313.47 | 43.87 | 314.61 | 5.41 | 311.75 | 3.91 |
| SR-2-20 | 274 | 130 | 0.47 | 0.0519 | 0.0012 | 0.3521 | 0.0078 | 0.0490 | 0.0005 | 282.03 | 50.84 | 306.27 | 5.88 | 308.62 | 3.06 |
| SR-2-23 | 362 | 156 | 0.43 | 0.0519 | 0.0019 | 0.3566 | 0.0109 | 0.0497 | 0.0008 | 279.92 | 82.48 | 309.64 | 8.19 | 312.56 | 5.20 |
| SR-2-24 | 662 | 292 | 0.44 | 0.0532 | 0.0014 | 0.3651 | 0.0103 | 0.0496 | 0.0007 | 336.08 | 58.20 | 316.04 | 7.67 | 312.09 | 4.30 |
| SR-2-25 | 331 | 138 | 0.42 | 0.0533 | 0.0011 | 0.3652 | 0.0077 | 0.0496 | 0.0005 | 340.20 | 45.72 | 316.10 | 5.71 | 312.00 | 3.16 |
| SR-2-26 | 337 | 136 | 0.40 | 0.0528 | 0.0014 | 0.3632 | 0.0100 | 0.0498 | 0.0005 | 321.75 | 59.55 | 314.57 | 7.45 | 313.06 | 3.26 |
| SR-2-27 | 373 | 141 | 0.38 | 0.0518 | 0.0015 | 0.3551 | 0.0096 | 0.0498 | 0.0006 | 277.32 | 67.20 | 308.56 | 7.18 | 313.25 | 3.80 |
| SR-2-28 | 431 | 193 | 0.45 | 0.0528 | 0.0011 | 0.3584 | 0.0082 | 0.0491 | 0.0006 | 320.75 | 49.17 | 311.01 | 6.11 | 309.00 | 3.97 |
| SR-2-29 | 425 | 173 | 0.41 | 0.0526 | 0.0017 | 0.3578 | 0.0116 | 0.0493 | 0.0006 | 313.00 | 73.37 | 310.54 | 8.67 | 310.16 | 3.92 |
| SR-1-01 | 316 | 117 | 0.37 | 0.0529 | 0.0013 | 0.3488 | 0.0089 | 0.0480 | 0.0005 | 324.23 | 57.73 | 303.78 | 6.67 | 302.25 | 3.03 |
| SR-1-02 | 197 | 73 | 0.37 | 0.0532 | 0.0013 | 0.3504 | 0.0089 | 0.0483 | 0.0006 | 337.88 | 56.82 | 305.04 | 6.70 | 303.94 | 3.73 |
| SR-1-03 | 388 | 184 | 0.47 | 0.0530 | 0.0011 | 0.3563 | 0.0074 | 0.0487 | 0.0006 | 330.41 | 45.73 | 309.42 | 5.57 | 306.76 | 3.45 |
| SR-1-04 | 435 | 232 | 0.53 | 0.0525 | 0.0011 | 0.3552 | 0.0094 | 0.0489 | 0.0006 | 306.14 | 49.76 | 308.59 | 7.04 | 307.91 | 3.51 |
| SR-1-07 | 1021 | 493 | 0.48 | 0.0531 | 0.0008 | 0.3646 | 0.0076 | 0.0495 | 0.0005 | 333.49 | 34.99 | 315.65 | 5.67 | 311.64 | 3.08 |
| SR-1-08 | 356 | 162 | 0.46 | 0.0531 | 0.0009 | 0.3637 | 0.0069 | 0.0497 | 0.0005 | 334.56 | 39.62 | 314.98 | 5.15 | 312.51 | 3.19 |
| SR-1-09 | 322 | 126 | 0.39 | 0.0535 | 0.0014 | 0.3608 | 0.0084 | 0.0492 | 0.0007 | 350.53 | 58.94 | 312.83 | 6.29 | 309.64 | 4.03 |
| SR-1-10 | 207 | 62 | 0.30 | 0.0543 | 0.0016 | 0.3624 | 0.0102 | 0.0485 | 0.0007 | 383.00 | 66.65 | 313.99 | 7.63 | 305.52 | 4.19 |
| SR-1-11 | 563 | 286 | 0.51 | 0.0526 | 0.0010 | 0.3553 | 0.0065 | 0.0490 | 0.0007 | 310.60 | 41.81 | 308.71 | 4.86 | 308.41 | 4.16 |
| SR-1-12 | 465 | 197 | 0.42 | 0.0518 | 0.0010 | 0.3481 | 0.0079 | 0.0486 | 0.0006 | 275.45 | 44.57 | 303.33 | 5.92 | 305.75 | 3.46 |
| SR-1-13 | 680 | 364 | 0.54 | 0.0520 | 0.0008 | 0.3543 | 0.0072 | 0.0491 | 0.0004 | 283.71 | 37.31 | 307.96 | 5.39 | 309.28 | 2.70 |
| SR-1-15 | 135 | 49 | 0.37 | 0.0521 | 0.0019 | 0.3487 | 0.0127 | 0.0488 | 0.0007 | 287.91 | 83.69 | 303.76 | 9.60 | 307.01 | 4.41 |
| SR-1-16 | 296 | 132 | 0.45 | 0.0526 | 0.0014 | 0.3548 | 0.0090 | 0.0491 | 0.0007 | 311.00 | 59.09 | 308.30 | 6.74 | 308.96 | 4.36 |
| SR-1-17 | 198 | 70 | 0.35 | 0.0533 | 0.0017 | 0.3595 | 0.0118 | 0.0492 | 0.0007 | 340.68 | 74.32 | 311.87 | 8.82 | 309.37 | 4.35 |
| SR-1-18 | 166 | 63 | 0.38 | 0.0532 | 0.0023 | 0.3562 | 0.0140 | 0.0486 | 0.0008 | 336.45 | 96.75 | 309.39 | 10.50 | 306.19 | 4.84 |
| SR-1-19 | 266 | 112 | 0.42 | 0.0524 | 0.0016 | 0.3553 | 0.0107 | 0.0492 | 0.0006 | 301.82 | 67.47 | 308.70 | 8.04 | 309.82 | 3.62 |
| SR-1-20 | 192 | 67 | 0.35 | 0.0518 | 0.0017 | 0.3505 | 0.0116 | 0.0490 | 0.0007 | 275.20 | 75.99 | 305.07 | 8.71 | 308.53 | 4.01 |
| SR-1-21 | 483 | 266 | 0.55 | 0.0532 | 0.0010 | 0.3608 | 0.0071 | 0.0492 | 0.0006 | 339.16 | 43.53 | 312.81 | 5.27 | 309.43 | 3.74 |
| SR-1-22 | 117 | 40 | 0.34 | 0.0526 | 0.0022 | 0.3584 | 0.0148 | 0.0498 | 0.0008 | 312.52 | 96.59 | 311.01 | 11.07 | 313.10 | 4.67 |
| SR-1-23 | 361 | 141 | 0.39 | 0.0515 | 0.0013 | 0.3503 | 0.0096 | 0.0493 | 0.0007 | 263.94 | 58.64 | 304.96 | 7.26 | 310.26 | 4.49 |
| SR-1-25 | 367 | 175 | 0.48 | 0.0526 | 0.0014 | 0.3546 | 0.0100 | 0.0487 | 0.0008 | 312.36 | 60.46 | 308.15 | 7.49 | 306.47 | 5.18 |
| SR-1-28 | 215 | 82 | 0.38 | 0.0529 | 0.0016 | 0.3540 | 0.0100 | 0.0488 | 0.0006 | 323.51 | 67.91 | 307.70 | 7.50 | 306.90 | 3.64 |
| SR-1-29 | 177 | 65 | 0.37 | 0.0511 | 0.0022 | 0.3438 | 0.0144 | 0.0488 | 0.0008 | 246.47 | 98.83 | 300.06 | 10.86 | 307.17 | 4.66 |
| SR-1-30 | 400 | 159 | 0.40 | 0.0530 | 0.0010 | 0.3591 | 0.0068 | 0.0491 | 0.0005 | 327.89 | 43.60 | 311.53 | 5.07 | 308.92 | 3.32 |

The results of zircons from the medium-fine-grained granites and granite porphyry yield a concordia diagram in Figure 7a,b, where the combined data points yield weighted mean $^{206}$Pb/$^{238}$U ages of 308.2 ± 1.5 Ma (MSWD = 0.68, n = 24) and 310.9 ± 1.5 Ma (MSWD = 0.22, n = 22), respectively. Therefore, we interpret the mean age as the crystallization age of the medium-fine-grained granites and granite porphyry. Both of them formed in the late Carboniferous epoch.

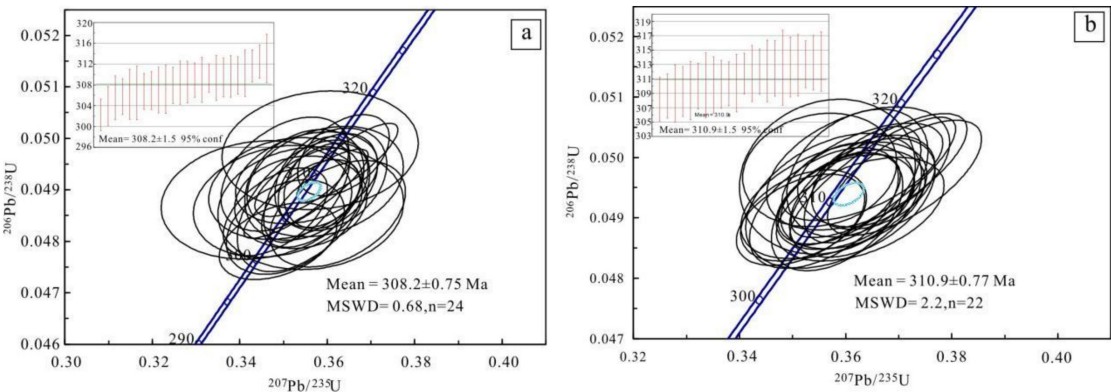

**Figure 7.** Zircon U–Pb concordia diagrams for (**a**) SR2, (**b**) SR1.

### 5.3. Zircon Hf Isotopes

The zircon Hf isotope compositions for the SR2 and SR1 are listed in Table 3. The $^{176}Lu/^{177}Hf$ ratios of all zircons are very close to or less than 0.002, indicating that after the formation of these zircons, the accumulation of radiogenic Hf was less, and the Hf isotope composition is reasonable [64]. The $^{176}Hf/^{177}Hf$ ratio can be used for geochemical tracing. The $^{176}Hf/^{177}Hf$ ratios of zircons from the SR2 vary between 0.282929 and 0.283032 (average 0.282974), with εHf(t) values ranging from 10.27 to 16.17 (average 13.71). Single model ages ($T_{DM1}$) and two-stage model ages ($T_{DM2}$) fall within the ranges of 319–535 Ma and 339–644 Ma, respectively. The $^{176}Hf/^{177}Hf$ ratios of zircons from the SR1 range from 0.282953 to 0.283033 (average 0.282972), with εHf(t) values ranging between 5.72 and 9.21 (average 7.08). Single model ages ($T_{DM1}$) and two-stage model ages ($T_{DM2}$) fall within the ranges 346~479 Ma and 309–557 Ma, respectively.

**Table 3.** Hf isotopic data of zircons from the SR2 and SR1 in the Sayashk Sn deposit.

| Spot no. | $t$ (Ma) | $\dfrac{^{176}Yb}{^{177}Hf}$ | $\dfrac{^{176}Lu}{^{177}Hf}$ | $\dfrac{^{176}Hf}{^{177}Hf}$ | 2σ | $\varepsilon_{Hf}(0)$ | $\varepsilon_{Hf}(t)$ | $T_{DM1}$ | $T_{DM2}$ | $f_{Lu/Hf}$ |
|---|---|---|---|---|---|---|---|---|---|---|
| SR-2-001 | 313 | 0.075538 | 0.001940 | 0.282967 | 0.000019 | 6.91 | 13.4 | 414 | 470 | −0.94 |
| SR-2-003 | 291 | 0.086482 | 0.002231 | 0.283016 | 0.000017 | 8.63 | 14.61 | 345 | 374 | −0.93 |
| SR-2-004 | 282 | 0.132216 | 0.003455 | 0.282905 | 0.000021 | 4.7 | 10.27 | 526 | 646 | −0.9 |
| SR-2-005 | 320 | 0.075191 | 0.001964 | 0.283024 | 0.000017 | 8.9 | 15.53 | 332 | 339 | −0.94 |
| SR-2-006 | 351 | 0.090929 | 0.002342 | 0.282929 | 0.000017 | 5.54 | 12.72 | 475 | 542 | −0.93 |
| SR-2-009 | 305 | 0.078083 | 0.002025 | 0.282967 | 0.000020 | 6.88 | 13.19 | 416 | 478 | −0.94 |
| SR-2-011 | 324 | 0.115128 | 0.002893 | 0.282893 | 0.000020 | 4.3 | 10.81 | 535 | 644 | −0.91 |
| SR-2-012 | 312 | 0.098395 | 0.002539 | 0.282984 | 0.000017 | 7.49 | 13.84 | 396 | 441 | −0.92 |
| SR-2-013 | 312 | 0.095046 | 0.002361 | 0.282990 | 0.000019 | 7.69 | 14.08 | 386 | 426 | −0.93 |
| SR-2-014 | 336 | 0.074679 | 0.001915 | 0.283032 | 0.000019 | 9.19 | 16.17 | 319 | 309 | −0.94 |
| SR-2-015 | 312 | 0.058124 | 0.001507 | 0.282993 | 0.000018 | 7.81 | 14.37 | 372 | 407 | −0.95 |
| SR-2-016 | 347 | 0.074915 | 0.001919 | 0.282986 | 0.000017 | 7.58 | 14.79 | 386 | 408 | −0.94 |
| SR-2-017 | 335 | 0.081711 | 0.002086 | 0.282989 | 0.000018 | 7.66 | 14.56 | 384 | 411 | −0.94 |
| SR-2-019 | 313 | 0.116336 | 0.002853 | 0.283004 | 0.000018 | 8.19 | 14.49 | 370 | 400 | −0.91 |
| SR-2-020 | 340 | 0.082512 | 0.002134 | 0.282937 | 0.000018 | 5.83 | 12.83 | 461 | 528 | −0.94 |
| DCRH-01 | 334 | 0.055608 | 0.001448 | 0.282967 | 0.000018 | 6.91 | 13.93 | 408 | 451 | −0.96 |
| DCRH-03 | 291 | 0.078398 | 0.002021 | 0.282964 | 0.000017 | 6.79 | 12.81 | 419 | 490 | −0.94 |
| DCRH-04 | 324 | 0.126125 | 0.003198 | 0.282934 | 0.000020 | 5.72 | 12.15 | 479 | 557 | −0.9 |
| DCRH-06 | 337 | 0.082700 | 0.002123 | 0.282996 | 0.000017 | 7.93 | 14.88 | 374 | 394 | −0.94 |
| DCRH-08 | 383 | 0.144140 | 0.003583 | 0.283033 | 0.000020 | 9.21 | 16.75 | 333 | 309 | −0.89 |
| DCRH-09 | 275 | 0.071345 | 0.001849 | 0.282987 | 0.000018 | 7.61 | 13.34 | 384 | 445 | −0.94 |
| DCRH-11 | 336 | 0.054312 | 0.001444 | 0.282962 | 0.000018 | 6.71 | 13.8 | 416 | 462 | −0.96 |
| DCRH-12 | 312 | 0.028669 | 0.000773 | 0.282943 | 0.000015 | 6.06 | 12.77 | 435 | 510 | −0.98 |
| DCRH-14 | 330 | 0.083431 | 0.002089 | 0.282993 | 0.000017 | 7.84 | 14.66 | 377 | 403 | −0.94 |
| DCRH-15 | 323 | 0.064451 | 0.001673 | 0.282964 | 0.000018 | 6.8 | 13.55 | 415 | 467 | −0.95 |
| DCRH-16 | 287 | 0.066353 | 0.001717 | 0.282951 | 0.000016 | 6.35 | 12.33 | 434 | 517 | −0.95 |
| DCRH-17 | 306 | 0.081286 | 0.002078 | 0.282970 | 0.000019 | 6.99 | 13.31 | 412 | 471 | −0.94 |
| DCRH-18 | 310 | 0.046680 | 0.001232 | 0.282953 | 0.000021 | 6.4 | 12.98 | 426 | 494 | −0.96 |
| DCRH-23 | 327 | 0.049078 | 0.001291 | 0.282967 | 0.000014 | 6.9 | 13.81 | 407 | 454 | −0.96 |
| DCRH-24 | 310 | 0.043277 | 0.001216 | 0.282958 | 0.000018 | 6.59 | 13.15 | 418 | 482 | −0.96 |
| DCRH-26 | 283 | 0.068651 | 0.001867 | 0.283014 | 0.000019 | 8.54 | 14.43 | 346 | 381 | −0.94 |

### 5.4. Molybdenite Re–Os Geochronology

The analytical results of molybdenite Re–Os dating are listed in Table 4. In seven molybdenite samples, Re = 70.50~462.0 ppb, Os = 0.0001~0.1658 ppb, $^{187}Re$ = 44.31~290.4 ppb and $^{187}Os$ = 0.2269~1.473 ppb. The model age of molybdenite is in the range of 306.5 Ma~298.1 Ma, with a weighted average age of 301.1 ± 3.1 Ma, MSWD = 2.0, isochron age of 301.4 ± 6.7 Ma, and MSWD = 3.4, which represents the metallogenic age of the Sayashk Sn deposit.

**Table 4.** Re–Os isotopic data for molybdenite from the Sayashk Sn deposit.

| Sample No. | Weight (g) | Re ng/g | | Os ng/g | | $^{187}$Re ng/g | | $^{187}$Os ng/g | | Model Age (Ma) | |
|---|---|---|---|---|---|---|---|---|---|---|---|
| | | Measured | 2σ | Measured | 2σ | Measured | 2σ | Measured | 2σ | Measured | 2σ |
| SR-6-3 | 0.10023 | 246.8 | 1.7 | 0.0111 | 0.0018 | 155.1 | 1.1 | 0.7724 | 0.0051 | 298.1 | 4.1 |
| SR-6-5 | 0.15002 | 168.3 | 1.6 | 0.0001 | 0.0008 | 105.8 | 1.0 | 0.5267 | 0.0037 | 298.1 | 4.5 |
| SR-6-6 | 0.10023 | 208.2 | 1.5 | 0.0001 | 0.0012 | 130.8 | 0.9 | 0.6551 | 0.0043 | 299.8 | 4.2 |
| SR-6-7 | 0.15043 | 134.2 | 1.6 | 0.0023 | 0.0008 | 84.37 | 1.00 | 0.4311 | 0.0031 | 305.9 | 5.2 |
| SR-6-8 | 0.10230 | 281.9 | 3.6 | 0.0011 | 0.0007 | 177.2 | 2.3 | 0.8874 | 0.0056 | 299.8 | 5.2 |
| SR-6-9 | 0.15023 | 462.0 | 4.1 | 0.0060 | 0.0005 | 290.4 | 2.6 | 1.473 | 0.009 | 303.7 | 4.4 |
| SR-6-10 | 0.10056 | 70.50 | 1.05 | 0.1658 | 0.0220 | 44.31 | 0.66 | 0.2269 | 0.0024 | 306.5 | 6.4 |

## 6. Discussion

### 6.1. Timing of Magmatism and Mineralization in the Sayashk Sn Deposit

The granites in the East Junggar region are mainly distributed along three major tectonic faults [65]. They could be divided into two cycles, Caledonian and Variscan. The Caledonian cycle was only developed in the late Caledonian period, such as the Yemaquan biotite granite, plagioclase-rich granite, and gneiss granite, which formed when the marginal sea basin at the end of the Silurian was closed and uplifted. They are orogenic uplift-type granites [66]. The Variscan cyclic granite can be divided into three stages: early Variscan, middle-late Variscan (340–290 Ma), and late Variscan (280–240 Ma). Among them, the early Warwick granite rock mass closely coexists with the ophiolite and is the final product of basic magmatic differentiation of the ophiolite. The lithology is mainly plagioclase granite and quartz diorite, which belong to pre-orogenic oceanic granite (M-type). The granites of middle-late Warwick and late Warwick are widely developed, and the rock types are mainly calc-alkali series and alkaline series. The calcium-alkali series lithology mainly includes pyroxene diorite, diorite, quartz diorite, monzolite, quartz monzolite, granodiorite, and plagiogranite. The alkaline series lithology is mainly quartz alkaline syenite, sodium-ferrite granite, sodium-amphibole granite, porphyry sodium-amphibole granite, and alkali-feldspar granite. Their formation was closely related to the collisional orogeny of East Junggar [32,65–67].

In the petrogenetic age, the La-ICP-MS zircon U–Pb ages of the SR2 and SR1 obtained in this study are 308.2 ± 1.5 Ma (MSWD = 0.52, n = 24) and 310.9 ± 1.5 Ma (MSWD = 0.21, n = 22), respectively. Regionally, Bai Jianke et al. (2018) carried out LA-ICP-MSICP-MS zircon U–Pb dating of the ore-bearing alkaline granites of the Huangyangshan graphite deposit in the East Junggar area and concluded that the ages of the ore-bearing granites of the No. 1 and No. 2 graphite ore bodies were 303.6 ± 4.0 Ma and 304.6 ± 3.7 Ma, respectively [68]. It was considered that the petrogenetic and metallogenic ages were both in the later part of the late Carboniferous; Ai et al. (2020) reported zircon U–Pb ages of 318.6 ± 4.2 Ma, 321.4 ± 2.7 Ma, and 305.1 ± 5.4 Ma for the medium-grained arfvedsonite, amphibole, and fine-grained biotite granites, respectively [69]. Sun et al. (2021) carried out U–Pb dating of LA-ICP-MS zircon for the Huangyangshan plutonic granite, medium-fine amphibolite granite, medium-grained biotite granite, and fine-grained biotite granite, and the results showed that the weighted average ages were 322.7 ± 4.5 Ma, 303.9 ± 4.0 Ma, 303.9 ± 2.1 Ma, and 301.1 ± 3.6 Ma, respectively [39]. Gan Lin et al. (2010) carried out LA-ICP-MS zircon U–Pb dating of granodiorite, monzogranite, and alkali-feldspar granite of the Yemaquan complex rock in eastern Junggar. The results are 304 ± 3.0 Ma, 300 ± 2.0 Ma and 297 ± 6.0 Ma, respectively, indicating that the Yemaquan rock was emplaced in the late Carboniferous and belonged to the product of magmatism in the post-collision stage of the eastern Junggar area [70]. Hu Wanlong (2016) carried out LA-ICP-MS zircon U–Pb dating of porphyritic granite, coarse-grained granite, and fine-grained granite in the Laoyaquan complex granite body in the eastern Junggar, and the results were 317.6 ± 3.3 Ma and 310.3 ± 3 Ma, respectively. Ma and 308.4 ± 2.8 Ma, formed in the late Carboniferous [71]; Li, Yuechen, etal, (2007) dated the zircon SHRIMP U–Pb of the Bellekuduk syenite

granite and obtained an age of 306 ± 5 Ma [72]. Nie Xiaoyong (2017) identified the early Carboniferous Qingshuidong plagioclase granite in the Eastern Junggar, with a harmonious age of 341.7 ± 1.7 Ma, quartz diorite with a harmonious age of 349.2 ± 6.7 Ma, and the Baijiigou granite has an age of 337.4 ± 5.8 Ma. In the East Junggar area, especially in the Kalamaili fault zone, magmatic activities are extensive and intense. With the characteristics of the complex rocks, it is not difficult to see from the above studies that these magmatic activities are characterized by multiple periods [73]. Han Baofu et al. (2006) believed that post-collisional plutonic magmatism occurred from 330–265 Ma and was concentrated from 330–310 Ma and 305–280 Ma [67]. Based on the extensive collection of previous research results (Table 5), this study considers that magmatic intrusions of 450–265 Ma were widely developed in this area, 420–410 Ma, 350–330 Ma, 320–300 Ma, and 300–280 Ma are four stages of relatively intense magmatic activity, especially 320–300 Ma, is the most intense period of magmatic activity (Figure 8).

**Table 5.** Typical intrusions and metal deposits of Kalamaili.

| Name | Age (Ma) | Testing Method | Testing Object | References |
|---|---|---|---|---|
| Seltek pluton | 314 ± 2 | U-Pb | Zircon | [65] |
| Ertai potassium feldspar granite | 319 ± 7 | SHRIMP | Zircon | [67] |
| Xiaohongshan pluton | 296 ± 4 | SHRIMP | Zircon | [67] |
| Sujiquan pluton | 295 ± 5 | SHRIMP | Zircon | [67] |
| Kamst pluton | 292 ± 7 | SHRIMP | Zircon | [67] |
| Belage Kuduk pluton | 273 ± 6 | SHRIMP | Zircon | [67] |
| Yebushan pluton | 268 ± 4 | SHRIMP | Zircon | [67] |
| Kubu Sunan pluton | 286 ± 3 | LA-ICP-MS | Zircon | [8] |
| Hilectic Harassu pluton | 381 ± 6 | SHRIMP | Zircon | [74] |
| Harassay pluton | 376 ± 10 | SHRIMP | Zircon | [74] |
| Bieliatun granite | 282 ± 5 | SHRIMP, LA-ICP-MS | Zircon | [59] |
| Sujiquan biotite granite | 304 ± 2 | LA-ICP-MS | Zircon | [59] |
| Huangyangshan pluton | 310 ± 4 | LA-ICP-MS | Zircon | [75] |
| Huangyangshan pluton | 302 ± 2 | LA-ICP-MS | Zircon | [75] |
| Sabei granite | 306 ± 3 | LA-ICP-MS | Zircon | [2] |
| Thorange Kuduk granite | 413 ± 8 | SHRIMP | Zircon | [72] |
| Belle Kuduk granite | 284 ± 5 | LA-ICP-MS | Zircon | [72] |
| Qiongheba pluton | 412.7 ± 3.3 | LA-ICP-MS | Zircon | [76] |
| Yemaquan pluton | 304 ± 3 | LA-ICP-MS | Zircon | [70] |
| Ulungu River Chakurtu pluton | 311.2 ± 2.5 | SHRIMP | Zircon | [77] |
| Ertai pluton | 279 ± 3 | SHRIMP | Zircon | [77] |
| Sayashk granite | 310.9 ± 1.5 | LA-ICP-MS | Zircon | This paper |
| Sayashk granite porphyry | 308.2 ± 1.5 | LA-ICP-MS | Zircon | This paper |
| Mutanyao granite | 349.8 ± 3.52 | LA-ICP-MS | Zircon | [78] |
| Basque granodiorite | 301 ± 2.5 | LA-ICP-MS | Zircon | [58] |
| Basque granodiorite | 310 ± 3.6 | LA-ICP-MS | Zircon | [58] |
| Nanmingshui gold deposit | 337.5 ± 3.9 | LA-ICP-MS | Sericite | [73] |
| Shuangquan gold deposit | 269 ± 9~260 ± 4 | Ar-Ar | Sericite | [79] |
| Shuangquan gold deposit | 310 | Ar-Ar | Inclusions in Quartz | [80] |
| Qingshui No. 48 old deposit | 311 ± 46 | Rb-Sr | Inclusions in Quartz | [81] |
| Jinshuiquan gold deposit | 271.7 ± 3.3 | Ar-Ar | Sericite | [81] |
| West of the Huangyangshan gold deposit | 318.4 ± 310.3 | LA-ICP-MS | Zircon | [82] |
| Sayashk Sn deposit | 301.1 ± 3.1 | Re-Os | Molybdenite | This paper |
| Sabei Sn deposit | 324.2 ± 3.4 | SHRIMP | Zircon | [31] |
| Ganliangzi Sn deposit | 314.0 ± 1 | Ar-Ar | muscovite | [83] |
| Kamust Sn deposit | 307.0 ± 1 | Ar-Ar | muscovite | [83] |

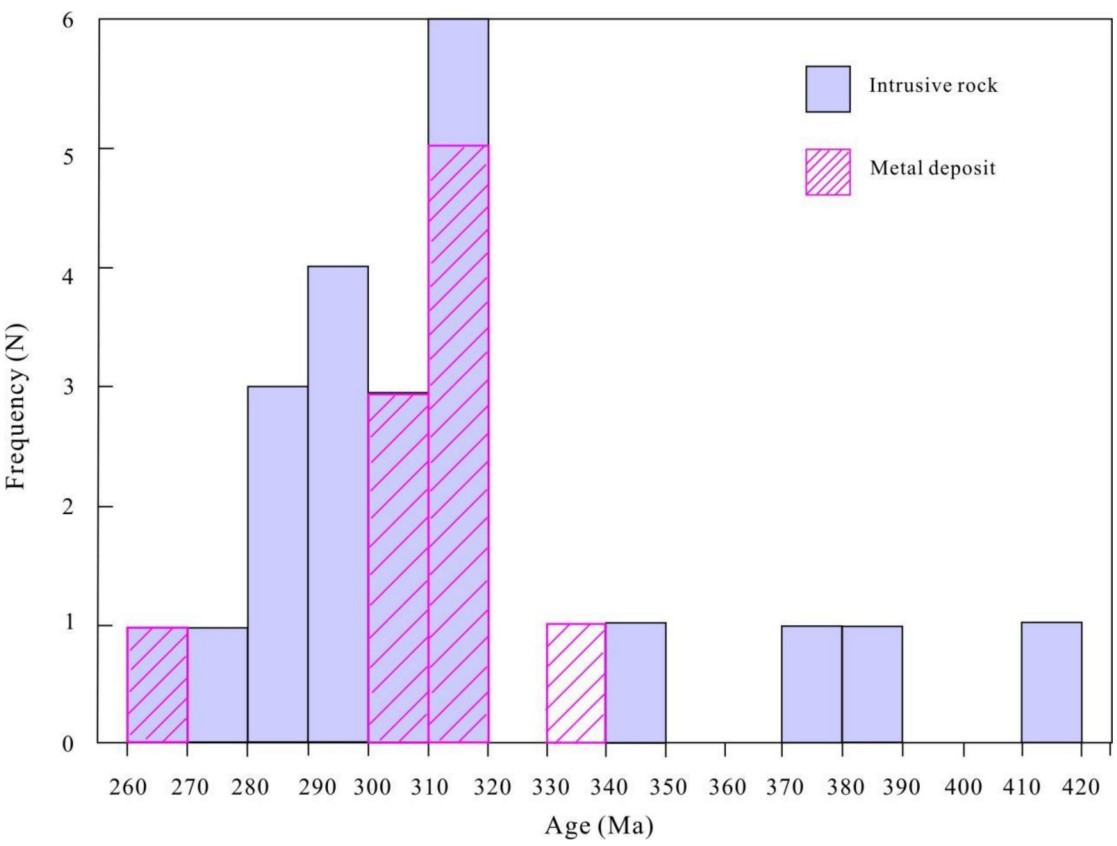

**Figure 8.** Age histogram of the typical intrusive rocks and some important metal deposits in the Kalamaili area.

In terms of metallogenetic epoch, a series of (tungsten) Sn deposits (points) and graphite deposits developed in the Belekuduk (tungsten) tin metallogenic belt in the East Junggar region, such as Kamster, Ganliangzi, Belekuduk, Sayashk, Sujiquan, and Hongtujingzi. In addition to the Huangyangshan and Sujiquan graphite deposits. They are mostly formed in the middle-late Carboniferous. For example, the $^{40}$Ar-$^{39}$Ar age of muscovite in the mica-type tin ore in the Bailekuduke Sn deposit is 306.8 ± 2.4 Ma~309.7 ± 2.4 Ma [3], $^{40}$Ar-$^{39}$Ar isotopic age of in quartz vein is 296.3 ± 2.6 Ma [6], Re–Os isotopic age of globular graphite in Huangyangshan graphite deposit is 332 ± 53 Ma [39], Rb-Sr isochron age of cassiterite-quartz vein whole rock of Ganliangzi Sn deposit is 307 ± 20 Ma and Rb-Sr isochron age of quartz fluid inclusions is 305 ± 25 Ma [41], Re–Os isochron age of molybdenite in tin ore of Sayashk Sn deposit is 307 ± 11 Ma [2], the model age of Molybdenite in the Sayashk Sn deposit is 298.1~306.5 Ma, with a weighted average age of 301.1 ± 3.1 Ma (MSWD = 2.0). The results show that the Sn deposits in this metallogenic belt were mainly concentrated between 320 Ma and 300 Ma (Figure 8). Because Sn deposits are strictly spatially controlled by alkaline granite, ore bodies of each Sn deposit occur in alkaline granite bodies and adjacent surrounding rocks, and tin metallogenesis is controlled by magma intrusion, differentiation, and gas–liquid alteration. Metallogenesis generally occurs in the late evolution of alkaline complex pluton and shows a good coupling relationship between diagenesis and mineralization in the late Carboniferous [32].

### 6.2. Rock Type, Petrogenesis and Nature of the Magma Source Area at the Sayashk Sn Deposit

6.2.1. Rock Type

As previously mentioned, the post-collisional plutonic magmatism in the East Junggar mainly occurred at 420–410 Ma, 350–330 Ma, 320–300 Ma, and 300–280 Ma. The late Silurian early Devonian granites are mainly found in the Jiangerkudu area and occur as batholiths or stocks. The lithology is mainly medium-grained biotite quartz monzodiorite, belonging

to the quasi-aluminous high potassium calc-alkaline rocks [76,84], with no dark inclusions found. The early Carboniferous granites are sporadically exposed south of the Qingshui-Sujiquan fault, and the rock types are mainly quartz diorite, oligoclase granite, granodiorite, quartz porphyry, and quartz syenite [4]; most of them are quasi-aluminum-peraluminous high potassium calc-alkaline rock series. Late Carboniferous granites are widely distributed north of the Kalamaili fault and are the main components of the alkali-rich granite belt in Kalamaili, mainly composed of Laoyaquan, Huangyangshan, Bellekuduk, Sujiquan, and Balebanaan. They are produced as batholith or plutons [5,31,75]. The Sayashk granites are formed on the north side of the Huangshan pluton. The SR2 and SR1 are characterized by high silicon and alkali contents and belong to the per-alkali rock series. The petrogenetic age of these plutons range from 320 Ma to 305 Ma, and the rock types include biotite granite, soda amphibole granite, alkali feldspar granite, and quartz alkaline syenite. In addition, dark inclusions are developed in most plutons (Figure 6), showing the characteristics of mixed crustal mantle sources.

SR2 and SR1 are mainly composed of quartz and alkali feldspar, silicon-rich (average value of SR1 is 78.65%), alkali-rich (SR1-average 9.60%, SR2-average 8.25%), calcite-poor (SR1-average 0.82%, SR2-average 0.22%), magnesium-poor (SR1-average 0.22%, SR2-average 0.07%). They are basically consistent with the major element characteristics of typical A-type granites locally and globally, with high $SiO_2$ average (73.35%~73.81%), alkaline-rich (average 8.42%~8.72%), and poor CaO (average 0.75%~0.82%) [85], enriched in HFSEs (Zr and Hf) and Ce, poor in Ba, Sr, Eu, P, and Ti, and show a typical right-dipping seagull-type REE distribution pattern, which conforms to the characteristics of A-type granite. Overall, in the case of high differentiation (aluminous A-type granite), I-type, S-type, and A-type granites are difficult to distinguish [86]. They often have the same mineralogical and geochemical characteristics [85]. Therefore, they still need to be distinguished.

The distinction between A-type granite and highly differentiated S-type granite is relatively simple; highly differentiated S-type granites have a higher content of $P_2O_5$ (average 0.14%) [87], and the content of $P_2O_5$ increases with increasing differentiation degree, but A-type granite has the opposite trend. SR2 and SR1 in this paper have low contents of $P_2O_5$ (SR2 average 0.03%, SR1—0.06%), and high content of $Na_2O$ (SR2 average 3.87%, SR1—4.63%). No characteristic minerals of S-type granite (such as primary muscovite, cordierite) are found, so it is impossible to be a highly differentiated S-type granite.

It is calculated that the zircon saturation temperatures of the two granites are 900 °C–908.2 °C (average 902.3 °C), and 836 °C–850.4 °C (average 843.3 °C), respectively, and the formation temperature is relatively high. The two plutons (TFeO) are 2.44–2.84 (average 2.68) and 1.01–1.27 (average 1.12), respectively. The total iron content of (TFeo) A-type granite is high, generally greater than 1%, while that of highly differentiated I-type granite is generally less than 1% [88], and it has high TFeo/(TFeo + MgO) values (average values are 0.925 and 0.938, respectively), showing the characteristics of A-type granite. Diagram of granite rock type discrimination (Nb-10,000 Ga/Al diagram and Ce-10,000 Ga/Al diagram) (Figure 9). The sample data plot in the A-type granite area. It is shown as A2-type granite in the classification diagram of the A-type granite subtype (Figure 10). Overall, the Sayashk granites are similar to the typical granite in the region from the geochemical characteristics of major elements, REE and trace elements, and so on (Figure 9) [9,30,59,70].

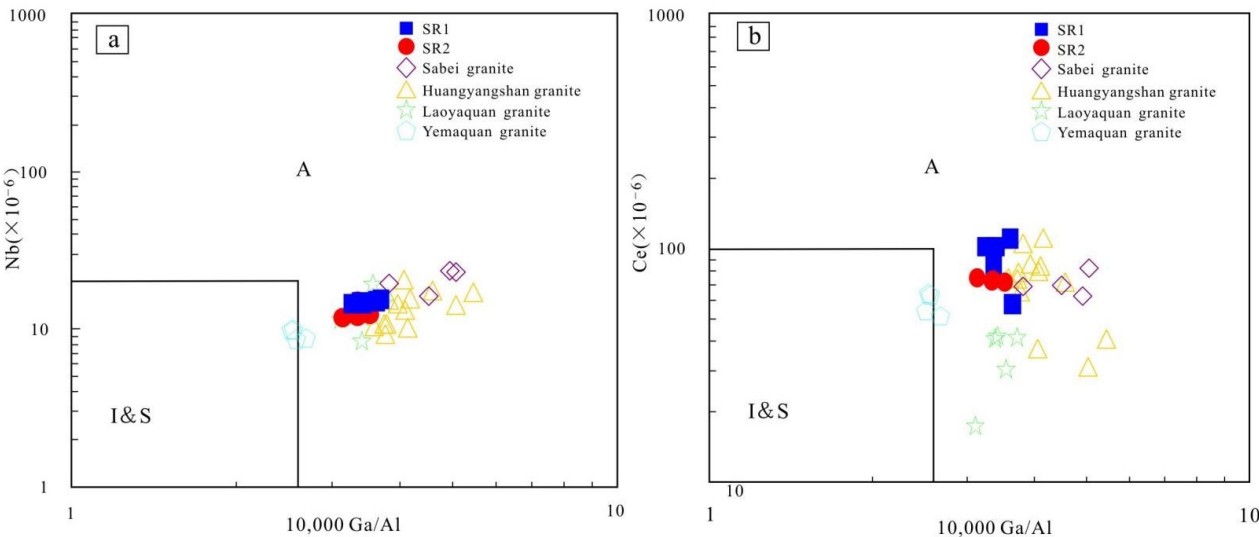

**Figure 9.** Nb-10,000 Ga/Al diagrams (**a**) and Ce-10,000 Ga/Al diagrams (**b**) for the medium-fine-grained granites and granite porphyry in the Sayashk Sn deposit. (After [86]). Sabei granite after [30], Huangyangshan granite after [75], Laoyaquan granite after [9], Yemaquan granite after [70].

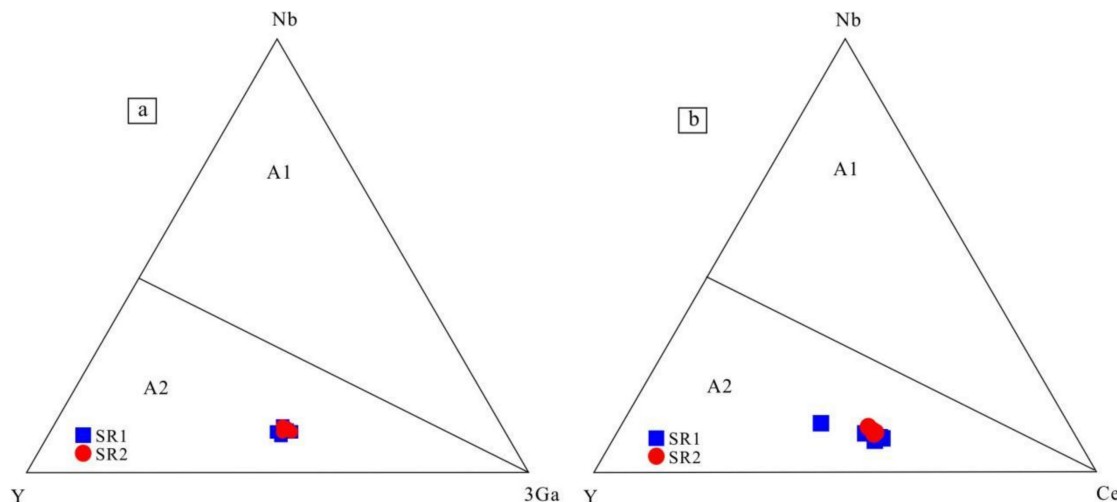

**Figure 10.** Y–Nb–3 Ga diagrams (**a**) and Y–Nb–Ce diagrams (**b**) for the medium-fine-grained granites and granite porphyry in the Sayashk Sn deposit (modified after [88]).

6.2.2. Petrogenesis and Nature of the Magma Source Area

At present, the main viewpoints on the petrogenetic types of A-type granite are as follows: (1) differentiation or partial melting of mantle-derived magma [89]; (2) mixing and melting of crustal mantle materials [90]; and (3) partial melting of crustal source material [91].

The acidic rocks produced by the crystal differentiation of mantle-derived alkaline basalt magmas show the characteristics of peralkaline, which is contradictory to the metaluminous or weakly peraluminous characteristics of the rocks in this paper, and the mantle-derived alkaline basalt magma produced. The temporal and spatial distribution of the acidic rocks is often related to a large number of intermediate-basic magmas, which is inconsistent with the geological facts of the widely developed acidic magmatic rocks in East Junggar, especially in the Huangyangshan area. Even in the entire northern Xinjiang region, the age of basic rocks is concentrated at approximately 280 Ma, indicating that these basic magmas should have ascended and emplaced after the formation of alkaline granites

in the East Junggar [92]. So, mantle-derived alkaline basalt magma is not the source rock of A-type granite in this area.

Previous studies showed that the abundance of Th in mantle-derived magma (0.05 ppm) was lower than that in a crustal source (16~21 ppm) [93]. SR2 and SR1 were 9.93~11.23 ppm and 11.76~11.23 ppm, respectively), which are larger than those in the mantle and close to those in the crust. The Nb/Ta values are 12.60~14.11 and 13.39~15.26, which are much lower than those in the mantle (Nb/Ta = 60) and closer to those in the crust (Nb/Ta = 11) [94], showing the characteristics of a mixed source in the crust and mantle.

Zircon has a strong closed system with a high Hf mass fraction and an extremely low $^{176}Lu/^{177}Hf$ value, and there is basically no radiogenic Hf accumulation after the system is closed. Therefore, zircon in situ Hf isotope research has become an important means to explain the crustal evolution and trace the magma source area.

Research shows that $\varepsilon Hf(t) > 0$ of zircon indicates that the magma came from the depleted mantle or partial melting of young crust that did not regenerate from the depleted mantle [95,96]. When $\varepsilon Hf(t)$ is close to the current mantle value, $\varepsilon Hf(t) < 0$ of zircon from granite indicates that the magma originated from the ancient crust and remelted. Zircon from two granites in the Sayashk Sn deposit $\varepsilon Hf(t)$ values of SR1 range from 10.27 to 16.17 (average at 13.71), $\varepsilon Hf(t)$ values of SR2 are between 5.72 and 9.21 (average 7.08), and single model ages ($T_{DM1}$) and two-stage model ages ($T_{DM2}$) of SR1 fall within the ranges of 319–535 Ma and 339–644 Ma. Single model ages ($T_{DM1}$) and two-stage model ages ($T_{DM2}$) of SR2 fall within the ranges of 346~479 Ma and 309~557 Ma. There is little difference between their two-stage model ages and zircon U–Pb ages, which indicates that the Sayashk granites may have originated from the partial melting of the young crust newly accreted from the depleted mantle. In the t–$\varepsilon Hf(t)$ diagram (Figure 11a), sample points plot above the Hf isotope evolution line of chondrites, characterized by enrichment. In the diagram of $t$–$^{176}Hf/^{177}Hf$ (Figure 11b), the sample points plot in the region between the chondrite Hf isotope evolution line and the depleted mantle, with enrichment characteristics. It is suggested that the Sayashk granites were the product of partial melting of juvenile crustal.

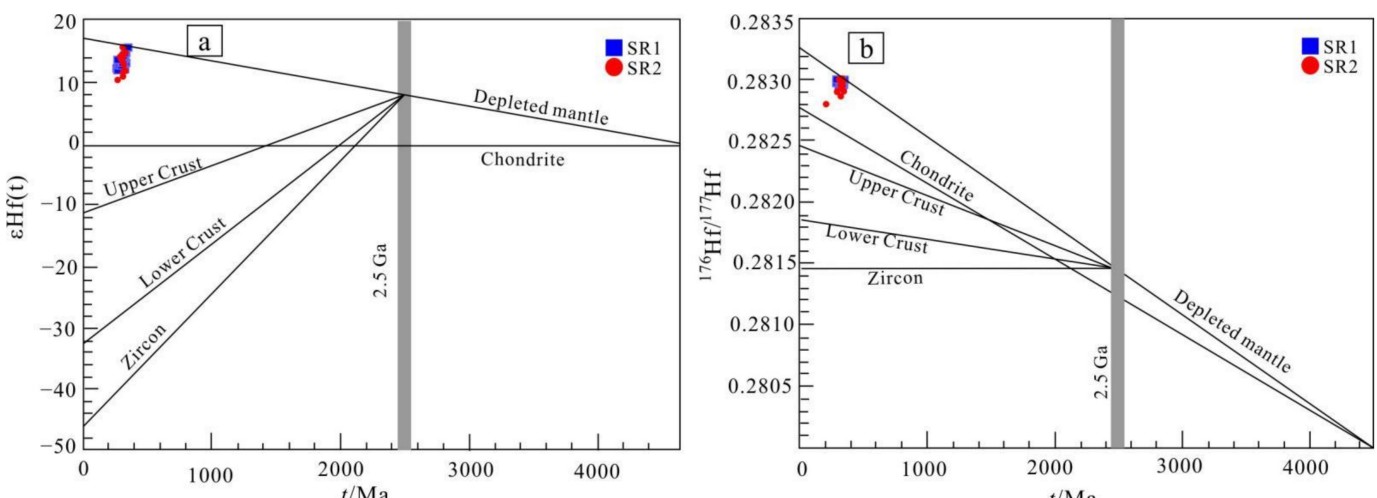

**Figure 11.** Zircon Hf isotopic compositions (**a**) and temporal variations (**b**) of the medium-fine-grained granites and granite porphyry in the Sayashk Sn deposit.

Previous studies have shown that A-type granites in the eastern Junggar region generally have high $\varepsilon Nd(t)$ values [2,59,97], which is consistent with the conclusion of this paper.

### 6.3. Tectonic Setting and Metallogenic Model

The Kalamaili area experienced multiple stages of ocean basin expansion, plate subduction, collisional orogeny, post-collision, and post-orogenic processes in the Paleozoic [98,99].

Intense tectonic movement accompanied by violent magmatism provided conditions for the development of gold and copper metallogenic series related to calc-alkaline granites and tin metallogenic series related to alkaline A-type granites [2,7,31,63].

Many studies have been performed on the magmatic rocks in this area according to the evolution of the tectonic environment. Yang et al. [8], believed that the later part of the late Carboniferous Huangyangshan alkaline granite was formed in the tectonic environment of a post-collisional extensional background. Wang [100] studied the early-late Carboniferous volcanic rocks in the Kalamaili area and concluded that the rocks were formed in the transitional environment of a post-collision and post-collision tensile environment. Chen et al. [101] studied the late Carboniferous volcanic rocks in this area and pointed out that the rocks had the dual characteristics of island arc volcanic rocks and a post-collision period. At the same time, they pointed out that the rocks did not form an island arc environment but a post-collision extension environment. The island arc characteristics indicate that magma contamination led to the inheritance of early island arc materials. Luo [102] studied the early late Carboniferous magmatic rocks in this area and showed that they were formed at the end of post-collision, marking the end of orogeny in the Kalamaili area. In conclusion, it is considered that the early-late Carboniferous in the Kalamaili area entered the end of collision activity, and the middle and late part of the late Carboniferous in the Kalamaili area were in a post-collision tension environment.

The Sayashk granitoids were formed during the latter part of the late Carboniferous. In the discrimination diagram of the granite tectonic environment, the granitic rocks in Sayashk are post-collisional granites (Figure 12a,b). Simultaneous $SiO_2$-FeOT/FeOT+MgO (Figure 13a) and $SiO_2$-$Al_2O_3$ diagrams (Figure 13b) diagrams display that the Sayashk granitic rocks are post-orogenic granites. This is consistent with the classification diagram of the A1–A2 subtype granite. This interpretation is also consistent with the consensus that the A-type granites in northern Xinjiang were formed in a post-collisional extensional environment [92].

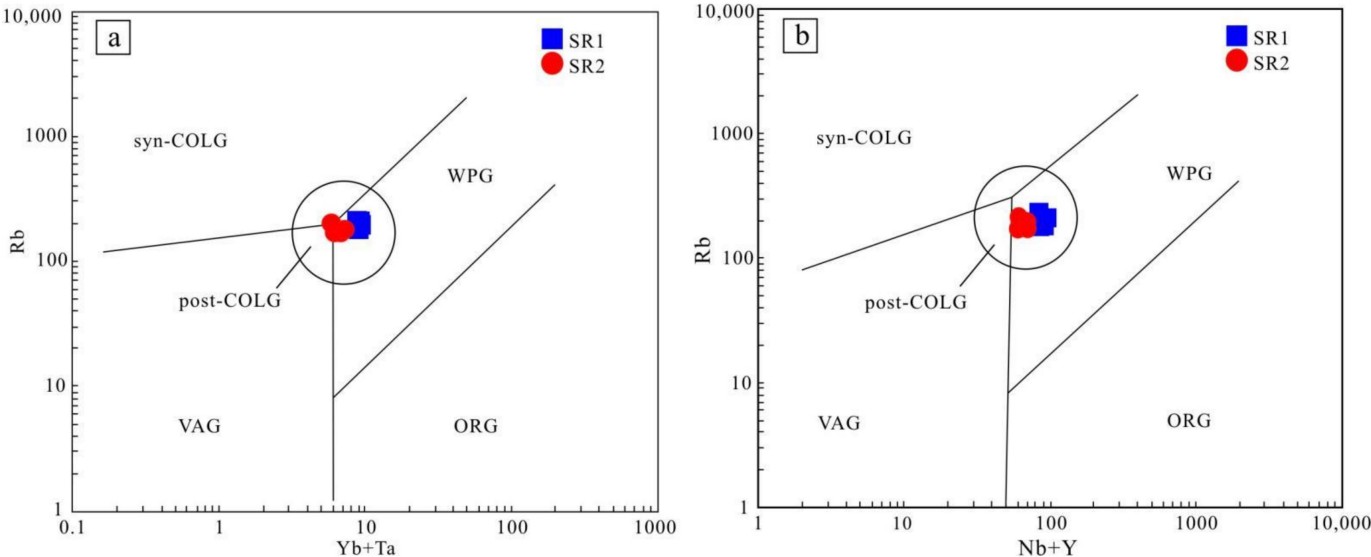

**Figure 12.** Tectonic environment discrimination diagram: (**a**) Yb-Ta diagram; (**b**) Y + Nb-Rb diagram (base drawing according to [103]). VAG: volcanic arc granite; ORG: ridge granite; WPG: intraplate granite; syn-COLG: collisional granite; post-COLG = post-collisional granites.

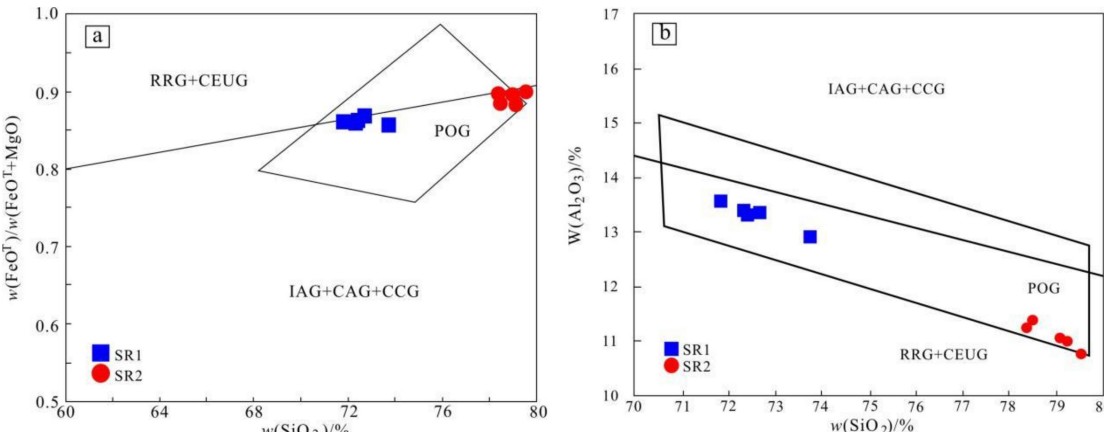

**Figure 13.** Tectonic environment discrimination diagram: (**a**) Yb–Ta diagram; (**b**) Y+Nb-Rb diagram (base drawing according to [104]). IAG: Island arc granite; CAG: continental arc granite; CCG: continental collision granite; POG: post-orogenic granite; RRG: rift-related granite; CEUG: continental uplift granite; OP: oceanic plagiogranite.

According to previous research results, the Paleozoic Kalamaili area can be roughly divided into five stages (Figure 14): ①: since the early Ordovician, the area has been in an environment of basin expansion [105]; ②: from the early Silurian to the late Silurian, the region formed a paleosubduction zone, the plate subducted southward, a strong fold orogeny occurred, and the uplift suffered denudation [106]; ③: in the early middle of the early Devonian, the Kalamaili ocean basin was opened, and the region was again in an extensional environment of extensional tectonics [99,106]; and ④: from the late part of the early Devonian to the early part of the late Carboniferous, the Kalamaili area changed from a continental margin facies to a mature island arc. At this stage, it was a strongly compressed island arc environment, and some magmatic rocks with a back-arc spreading environment were formed [2,74]. ⑤: After the late Carboniferous, the Kalamaili area entered the post-orogenic stage and once again changed from a compressive environment to an extensional environment [3,7,9,41]. At the same time, it is also the most frequent stage of magmatic activity in this area, forming many representative magmatic rocks, such as the Huangyangshan pluton, Sabei pluton and Kubusunan pluton.

The ore-forming material of the Kalamaili Sn ore belt comes from granite. The ore-forming hydrothermal solution is mainly late magmatic hydrothermal solution, and a small amount of atmospheric water is mixed later.

During the emplacement process of the original Sn-rich magma, with the progress of magma evolution, elements incompatible with early minerals such as Sn were gradually enriched and separated in late residual magmas and hydrothermal fluids.

Late ore-bearing (Sn) hydrothermal fluids with high contents of silicon, alkalis, and volatiles (mainly F and Cl) were gradually separated from the residual magma. The granites on both sides of the fault zone underwent alteration and alkali metasomatization (potassium feldsparization and albite). Under high-temperature conditions, mineralization alteration continues, and sand graining and silicification occur. The Sn element in the ore-bearing hydrothermal fluid continuously precipitated, and finally, a quartz vein-type tin ore body formed in a favorable position. The Sayashk Sn deposit is an Sn deposit formed in an extensional environment after the late Carboniferous.

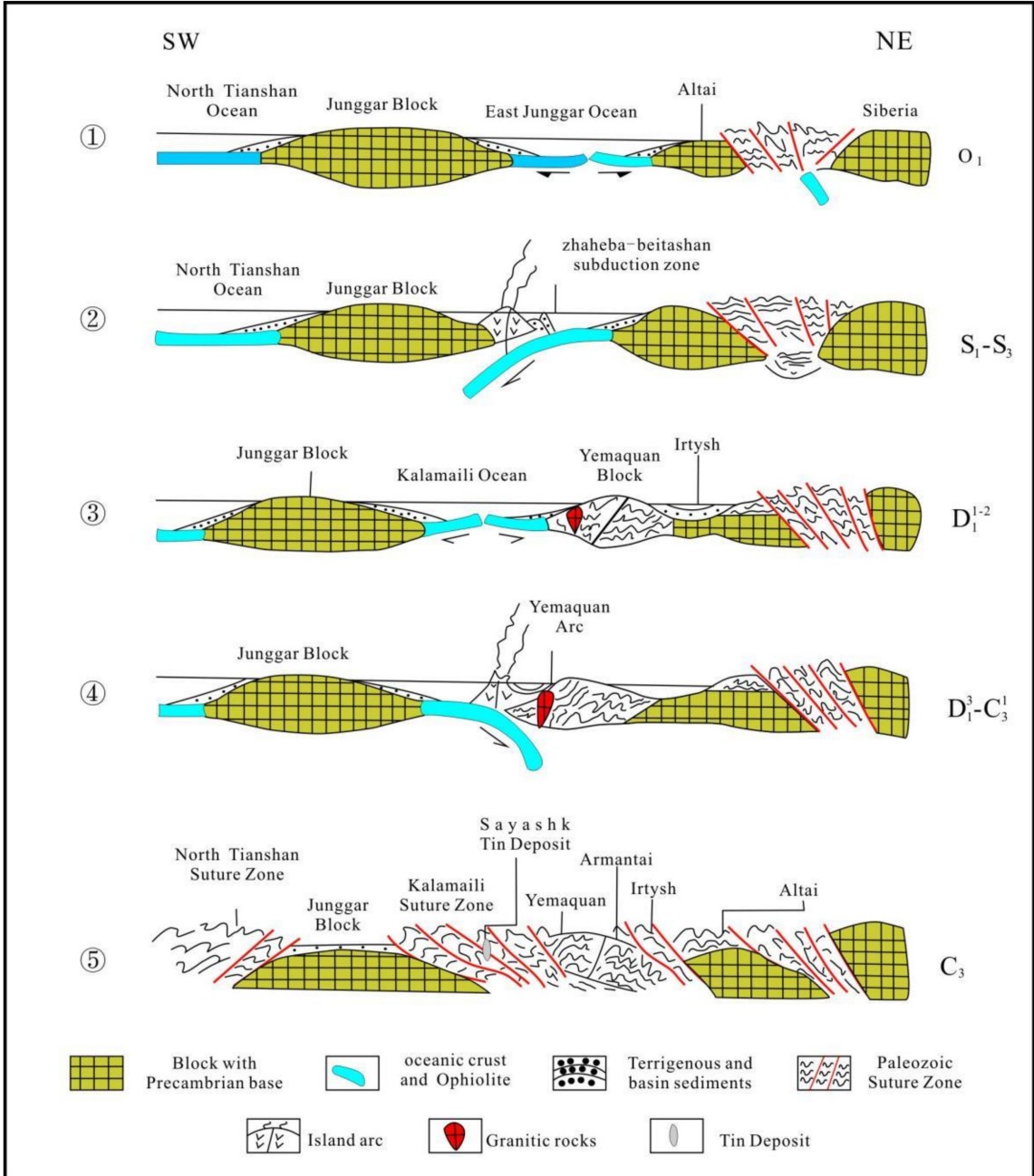

**Figure 14.** Diagram of Paleozoic tectonic evolution in East Junggar [106].

## 7. Conclusions

1.  The Sayashk Sn deposit is spatially, temporally, and genetically closely related to the granite porphyry and the medium-fine-grained granite. Both zircon U–Pb ages are $308.2 \pm 1.5$ Ma and $310.9 \pm 1.5$ Ma, respectively. The molybdenite isochron age is $301.4 \pm 6.7$ Ma, which represents the metallogenic age of the Sayashk Sn deposit. All of them formed in the late Carboniferous epoch.
2.  The medium-fine-grained granites (SR2) and granite porphyry (SR1) are rich in Si, rich alkali, poor Ca, poor Mg, enrichment HFSE (Zr, Hf) and Ce, loss Ba, Sr, Eu, P, Ti, which belongs to typical A-type granite. It shows that the mixed crustal mantle source is derived from the partial melting of juvenile crustal.

3.  The Sayashk Sn deposit was formed after the late Carboniferous, and the Kalamaili area entered the post-orogenic stage, which was formed from a compressive environment to an extensional environment.

**Author Contributions:** Conceptualization, Z.S. and G.L.; methodology, Y.R.; software, X.C.; investigation, Z.S., C.W. and X.S. resources, Z.L.; data curation, X.S.; writing—original draft preparation, Z.S.; writing—review and editing, Y.R. All authors have read and agreed to the published version of the manuscript.

**Funding:** This research was funded by the Fundamental Research Funds for the Central Universities of China (Grant Number ZY20215112) and the Science and Technology Research Project of Colleges and Universities in Hebei Province (Grant Number ZD2021405).

**Institutional Review Board Statement:** Not applicable.

**Informed Consent Statement:** Not applicable.

**Data Availability Statement:** Not applicable.

**Acknowledgments:** We sincerely appreciate the detailed and constructive reviews and suggestions from two anonymous reviewers, which greatly improved this paper.

**Conflicts of Interest:** The authors declare no conflict of interest.

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
