# Peer review of "Age, Genesis and Tectonic Setting of the Sayashk Tin Deposit in the East Junggar Region: Constraints from Lu–Hf Isotopes, Zircon U–Pb and Molybdenite Re–Os Dating"

_minerals, doi:10.3390/min12091063_

Round 1

Reviewer 1 Report

This manuscript contains a lot of what appear to be excellent data, and the conclusions are generally consistent with these data. However, it is poorly written so that many of the arguments are lost in the narrative. Most of them are minor and/or fairly easy to fix, they distract the science content of the manuscript. Instead of re-writing all the passages, which is really unfair to reviewers, I underlined most (but not all) of them. Examples are detailed in the following:

A) I am unfamiliar with some of the words used.  For example, what is meant by “independent tin deposit”, “deposits (points) distributed”, “rock masses”, “dense or sparse (star) point”, ”thin film” …….?

B) Some of the words are literally converted from the original dialect to English. For example, “at home and abroad” should be “locally and globally/universally”?

C) “Tin” is sometimes called “Sn” – use “tin (Sn)” in the beginning and use “Sn” thereafter, unless at the beginning of the sentence. That is, acronyms (or symbols) are usually not used at the beginning of the sentence. Do the same for other elements mentioned in the text.

D) There are many spelling errors – although correct sometimes: e.g., Sayashk, Karamaili, etc. are spelled differently in many places. 

Some of the isotopes are not written properly (# superscripts are not consistent).

E) “Diagenetic” age is not the same as “genetic” age – the latter is what is meant in this paper.

F) “Etc.” can be used informally (like here in the review), but should not be used in a scientific paper.

G) “Porphyry granite (SR1)” and “medium- to fine-grained grained granite (SR2)” should be defined in the beginning of the discussion and, then, SR1 and SR2 should be used when these rock units are needed to be mentioned again, unless at the beginning of the sentence.

H) Most of the analytical values are reported in scientific notations (i.e., ###  X  10-#). Typically, these are reported as %, ppm, ppb, ppt.

I) Some specific suggestions: 

Abstract: 

……….

To constrain the age, genesis, and tectonic setting of the Sayashk tin deposit in the East Jungar region, we conducted bulk-rock geochemical analysis of the granite porphyry (SR1) and medium- to fine-grained granite (SR2) hosts (??) of the deposit, LA-ICP–MS zircon U–Pb dating and Lu-Hf isotopic analysis, as well as molybdenite Re-OS dating and combined our results with the metallogenic conditions and other geological characteristics of the deposit. The results show that the Sayashk tin deposit is indeed spatially, temporally, …..  …………..    diagenetic age.

L131: 3. Geology of the ore deposit

L235-241: Do not repeat data that are presented in the tables. Simply summarize them and/or report only the averages. (There are a few instances like this in the manuscript – e.g., L430-435.)

L247-249: Since Ritman index is not defined, simply state something like “As the samples have high SiO2 contents (>70 wt%), we used ……”

L259-262: Something is wrong with the sentence: Repeat - something like “LREE/HREE values, represented by (La/Yb)N ratios, range from xx-xx for SR1 …….”

L327-331: This paragraph should be part of the introduction and/or regional geology section(s). Avoid redundancy.

L484-485: What does “crystallization and differentiation” mean? Crystallization is a possible differentiation mechanism; thus, I assume it means fractional crystallization? 

J) Major science errors:

Some of the isotopic values, especially for Hf isotopes, are not reported consistently – sometimes 6 decimal places, sometimes up to 8 decimal places – which are way beyond the analytical uncertainties.

L216-219: The reported analyses for the standard zircon are not similar (and not the same numbers)! Thus, the analytical results here are not directly comparable to literature data.

The crystallization and metallogenesis ages reported here are consistently ~300 Ma. However, the conclusion also says that the mixed crustal source of the magmas and metals (?) consists of newly-formed (Cenozoic, ≤ 66 Ma) crustal materials (e.g., see abstract and text).

How can you produce ~300 Ma granites and metals from a ~66 Ma crust?

Author Response

Response to Reviewer 1 Comments

Point 1: I am unfamiliar with some of the words used. For example, what is meant by “independent tin deposit”, “deposits (points) distributed”, “rock masses”, “dense or sparse (star) point”, “thin film” …….? 

Response 1: “independent tin deposit”:The original meaning is the tin-dominated or only tin-forming deposit. “independent tin deposit”has been changed to “dominated by Sn”.

“deposits (points) distributed”:The original sentence is”There are a number of tin deposits (points) distributed from west to east”. I don't think there is a problem.

“rock masses”has been changed to “pluton”.

“dense or sparse (star) point”、“thin film”:The original text has been revised to“The cassiterite and molybdenite is distributed in a dense star point, malachitization is distributed in a thin film.”

The above changes have also been made in other similar places in the original text.

Point 2: Some of the words are literally converted from the original dialect to English. For example, “at home and abroad” should be “locally and globally/universally”?

Response 2: “at home and abroad” has been changed to “locally and globally”.

“Sn mineralization of a series of slightly alkaline granite porphyry small bodies” has been changed to “tin mineralization of a series of slightly alkaline granite porphyry”

“complex intrusive rock bodies “ has been changed to “complex intrusive rocks“.

“”around the world“ has been changed to “all over the world”.

“predecessor” has been changed to “Many researchers ”or “a lot of researchers”.

Point 3: “Tin” is sometimes called “Sn”– use “tin (Sn)” in the beginning and use “Sn” thereafter, unless at the beginning of the sentence. That is, acronyms (or symbols) are usually not used at the beginning of the sentence. Do the same for other elements mentioned in the text.

Response 3: I have modified the full text according to the suggestion,and annotated it in red.

Point 4: .There are many spelling errors – although correct sometimes: e.g., Sayashk, Karamaili, etc. are spelled differently in many places.

Some of the isotopes are not written properly (# superscripts are not consistent).

Response 4: Isotope writing has been revised correctly. For example: 206Pb/238U , SiO2 ,176Hf/177Hf.

Point 5: .“Diagenetic” age is not the same as “genetic” age – the latter is what is meant in this paper.

Response 5: “Diagenetic” age is not the same as “genetic” age, but I think the “age”should be “diagenetic age”, and most similar articles choose “diagenetic age”. So, I didn ' t make any changes.

Point 6: .“Etc.” can be used informally (like here in the review), but should not be used in a scientific paper.

Response 6: “Etc.”has been changed to “and so on”,and annotated it in red.

Point 7: “Porphyry granite (SR1)” and “medium- to fine-grained grained granite (SR2)” should be defined in the beginning of the discussion and, then, SR1 and SR2 should be used when these rock units are needed to be mentioned again, unless at the beginning of the sentence.

Response 7: I have modified the full text according to the suggestion,and annotated it in red.

Point 8: Most of the analytical values are reported in scientific notations (i.e., ###  X  10-#). Typically, these are reported as %, ppm, ppb, ppt.

Response 8: I have revised the analytical values in the manuscript as suggested by the reviewer.

Point 9: To constrain the age, genesis, and tectonic setting of the Sayashk tin deposit in the East Jungar region, we conducted bulk-rock geochemical analysis of the granite porphyry (SR1) and medium- to fine-grained granite (SR2) hosts of the deposit, LA-ICP–MS zircon U–Pb dating and Lu-Hf isotopic analysis, as well as molybdenite Re-OS dating and combined our results with the metallogenic conditions and other geological characteristics of the deposit. The results show that the Sayashk tin deposit is indeed spatially, temporally, …..  …………..    diagenetic age.

Response 9: Thank the reviewer for helping to improve the quality of the manuscript, and the manuscript has been revised according to the reviewer's suggestions.

Point 10: L131: 3. Geology of the ore deposit

Response 10: Thank the reviewer for helping to improve the quality of the manuscript, and the manuscript has been revised according to the reviewer's suggestions.

Point 11: L235-241: Do not repeat data that are presented in the tables. Simply summarize them and/or report only the averages. (There are a few instances like this in the manuscript – e.g., L430-435.)

Response 11: Thank the reviewer for helping to improve the quality of the manuscript, and the manuscript has been revised according to the reviewer's suggestions.

Point 12: L247-249: Since Ritman index is not defined, simply state something like “As the samples have high SiO2 contents (>70 wt%), we used ……”

Response 12: Thank the reviewer for helping to improve the quality of the manuscript, and the manuscript has been revised according to the reviewer's suggestions.

Point 13: L259-262: Something is wrong with the sentence: Repeat - something like “LREE/HREE values, represented by (La/Yb)N ratios, range from xx-xx for SR1 …….”

Response 13: Thank the reviewer for helping to improve the quality of the manuscript, and the manuscript has been revised according to the reviewer's suggestions.

Point 14: L327-331: This paragraph should be part of the introduction and/or regional geology section(s). Avoid redundancy.

Response 14: I have deleted this section, regional geology has been introduced similarly.

Point 15: L484-485: What does “crystallization and differentiation” mean? Crystallization is a possible differentiation mechanism; thus, I assume it means fractional crystallization?

Response 15: I changed it to “The acidic rocks produced by the crystal differentiation of mantle-derived alkaline basalt magmas show the characteristics of peralkaline,”

Point 16:  Some of the isotopic values, especially for Hf isotopes, are not reported consistently – sometimes 6 decimal places, sometimes up to 8 decimal places – which are way beyond the analytical uncertainties.

Response 16: I have revised according to the reviewer's comments.

Point 17: L216-219: The reported analyses for the standard zircon are not similar (and not the same numbers)! Thus, the analytical results here are not directly comparable to literature data.

Response 17: I have revised according to the reviewer's comments.

Point 18: The crystallization and metallogenesis ages reported here are consistently ~300 Ma. However, the conclusion also says that the mixed crustal source of the magmas and metals (?) consists of newly-formed (Cenozoic, ≤ 66 Ma) crustal materials (e.g., see abstract and text).How can you produce ~300 Ma granites and metals from a ~66 Ma crust?

Response 18: The original sentence was translated incorrectly. I have modified it to “indicating that the Sayashk granite may be the product of partial melting of juvenile crustal. “

I have revised all the questions marked by the reviewer in the PDF one by one.

Reviewer 2 Report

This is an interesting paper that illustrate that the Sayashk tin deposit is spatially, temporally, and genetically related closely to the granitic porphyry and the medium-fine-grained granite by usinggeochemical element analysis, LA-ICPMS zircon UPb dating, LU-Hf isotopic composition and molybdenite Re-Os dating.The study will benefit geologists who expertise in research on the relationship between granites and tin deposits in the Kalamaili alkaline granite belt. It is suitable for publication after a minor revision. The revised paper should address the following questions:

(1) Supply Re-Os isotopic age before” Combined with previous research results,” in line 35. 

(2) Cassiterite can be used for dating to exactly establish ore-forming age.

3In the A/CNK versus A/NK diagram, all samples are “metaluminous in Figure 4c but not “peraluminous” mentioned in line 251 in text.

(4) Line 270: showing the characteristics of typical A-type granite should be changed into suggesting dominant plagioclase fractionation in more evolved melts (Nandedkar et al., 2016).

Some revised suggestions can be found in the attached file.

Author Response

Response to Reviewer 2 Comments

Point 1: Supply Re-Os isotopic age before” Combined with previous research results,” in line 35.

Response 1: The isochron age of molybdenite is 301.4±6.7 Ma,it was mentioned in line 24.

Point 2: Cassiterite can be used for dating to exactly establish ore-forming age.

Response 2: The U-Pb age of cassiterite was designed when the problem of metallogenic chronology was solved. Unfortunately, there were too few samples to select enough cassiterite.

Point 3: In the A/CNK versus A/NK diagram, all samples are “metaluminous” in Figure 4c but not “peraluminous” mentioned in line 251 in text.

Response 3: It was Modified. In the A/CNK versus A/NK diagram, all samples are metaluminous (Figure 4c).

Point 4: Line 270: “showing the characteristics of typical A-type granite” should be changed into “suggesting dominant plagioclase fractionation in more evolved melts (Nandedkar et al., 2016).”

Response 4: Thank the reviewer for helping to improve the quality of the manuscript, and the manuscript has been revised according to the reviewer's suggestions.

Point 5: Some revised suggestions can be found in the attached file.

Response 5: I have revised all the questions marked by the reviewer in the PDF one by one.

Round 2

Reviewer 1 Report

I thank the authors for their positive responses to almost all my comments/suggestions. However, the English language still needs to be improved. Here are a few examples:

L14-15: The Sayashk tin (Sn) deposit … Xinjiang Province and forms part of the Kalamaili 

L26-27: granites and granite porphyries are characteristically rich in Si and alkali, poor in Ca and Mg, rich in high field-strength elements (HFSE, e.g., Zr, Hf) and Ce, and deficient in Ba, Sr, Eu, P and Ti. 

L43-44: The Beilekuduk tin (Sn) ore belt …. first metallogenic belt dominated by Sn found

L57-58: …. alkali feldspar granite, and other granitic rocks, and most of them belong to highly differentiated ……

L113: all of which occur as batholiths. 

“Etc.” was replaced by “and so on” – this is not what I meant. What I meant is that it is not scientific to write “etc.”, “and so on” and the like. It should be replaced by, for example, “and other granitic rocks” if the previous subjects are granitic rocks, “and other characteristic  petrographic features” if the previous subjects are petrographic features, and “and other locations” if the previous subjects are locations. 

I am sure that the authors mean petrogenetic age (or age of rock formation and/or crystallization). Diagenetic age means the time when sediments are cemented together to form sedimentary rocks. It does not matter if diagenetic age had been used before, it is petrogenetic age in this manuscript.

My biggest concern is still the Hf isotopes:

In L213-214 it says: During analyses, the 176Hf/177Hf and 176Lu/177Hf ratios of the standard zircon (91500) were 0.282504±20 (2σ; n = 12) and 0.00021; the former is similar to the commonly accepted 176Hf/177Hf ratios of 0.282304±20 (2σ) measured using the solution method 

Unless there is a typing error (please check all the numbers), my understanding is that the values are clearly not similar – the difference is way outside the analytical uncertainties of 0.0000020. This means the Hf isotopes and all related data for the samples are systematically higher than published data that are used for comparison and interpretation. This has to be corrected. One possible way that is being used in cases like this is to re-normalize the average lab standard results to the accepted results. In this case, the accepted value is 0.282304, and the lab result is higher, at 0.282504, meaning the lab is systematically producing 176Hf/177Hf ratios that are 0.000200 higher than what the rest of the community is producing. 

A way to remedy the situation is to re-normalize the lab standard result to the accepted value. Then, all lab results should all be re-normalized also – a possible way is a simple substraction (or a more sophisticated per mass unit renormalization); then results can be reported, plotted and interpreted. The re-normalization of Hf isotope values and how it is done to the accepted value need to be explained in notes under appropriate report table(s).

Author Response

Response to Reviewer 1 Comments

Point 1: L14-15: The Sayashk tin (Sn) deposit … Xinjiang Province and forms part of the Kalamaili

Response 1: Thanks the reviewer for helping to improve the quality of the manuscript, and the manuscript has been revised according to the reviewer's suggestions.

Point 2: SL26-27: granites and granite porphyries are characteristically rich in Si and alkali, poor in Ca and Mg, rich in high field-strength elements (HFSE, e.g., Zr, Hf) and Ce, and deficient in Ba, Sr, Eu, P and Ti.

Response 2: Thanks the reviewer for helping to improve the quality of the manuscript, and the manuscript has been revised according to the reviewer's suggestions.

Point 3: L43-44: The Beilekuduk tin (Sn) ore belt …. first metallogenic belt dominated by Sn found.

Response 3: Thanks the reviewer for helping to improve the quality of the manuscript, and the manuscript has been revised according to the reviewer's suggestions.

Point 4: L57-58: …. alkali feldspar granite, and other granitic rocks, and most of them belong to highly differentiated ……

Response 4: Thanks the reviewer for helping to improve the quality of the manuscript, and the manuscript has been revised according to the reviewer's suggestions.

Point 5: L113: all of which occur as batholiths.

Response 5: Thanks the reviewer for helping to improve the quality of the manuscript, and the manuscript has been revised according to the reviewer's suggestions.

Point 6: “Etc.” was replaced by “and so on” – this is not what I meant. What I meant is that it is not scientific to write “etc.”, “and so on” and the like. It should be replaced by, for example, “and other granitic rocks” if the previous subjects are granitic rocks, “and other characteristic  petrographic features” if the previous subjects are petrographic features, and “and other locations” if the previous subjects are locations.

Response 6: “and so on” in the L137, L335, L337,L410,L415,L419,L565 have been revised.

Point 6: I am sure that the authors mean petrogenetic age (or age of rock formation and/or crystallization). Diagenetic age means the time when sediments are cemented together to form sedimentary rocks. It does not matter if diagenetic age had been used before, it is petrogenetic age in this manuscript.

Response 6: “diagenetic age” has been changed into “petrogenetic age”.

Point 7: My biggest concern is still the Hf isotopes:

In L213-214 it says: During analyses, the 176Hf/177Hf and 176Lu/177Hf ratios of the standard zircon (91500) were 0.282504±20 (2σ; n = 12) and 0.00021; the former is similar to the commonly accepted 176Hf/177Hf ratios of 0.282304±20 (2σ) measured using the solution method

Unless there is a typing error (please check all the numbers), my understanding is that the values are clearly not similar – the difference is way outside the analytical uncertainties of 0.0000020. This means the Hf isotopes and all related data for the samples are systematically higher than published data that are used for comparison and interpretation. This has to be corrected. One possible way that is being used in cases like this is to re-normalize the average lab standard results to the accepted results. In this case, the accepted value is 0.282304, and the lab result is higher, at 0.282504, meaning the lab is systematically producing 176Hf/177Hf ratios that are 0.000200 higher than what the rest of the community is producing.

A way to remedy the situation is to re-normalize the lab standard result to the accepted value. Then, all lab results should all be re-normalized also – a possible way is a simple substraction (or a more sophisticated per mass unit renormalization); then results can be reported, plotted and interpreted. The re-normalization of Hf isotope values and how it is done to the accepted value need to be explained in notes under appropriate report table(s).

Response 7:  I have contacted the laboratory to determine the test process and data results of Hf isotopes, and corrected it.
